# GAIA: A Data Flywheel System for Training GUI Test-Time Scaling Critic Models

## Abstract

While Large Vision-Language Models (LVLMs) have significantly advanced GUI agents' capabilities in parsing textual instructions, interpreting screen content, and executing tasks, a critical challenge persists: the irreversibility of agent operations—where a single erroneous action can trigger catastrophic deviations. To address this, we propose the **G**UI **A**ction Crit**i**c's Dat**a** Flywheel System (GAIA), a training framework that enables the models to have iterative critic capabilities, which are used to improve the Test-Time Scaling (TTS) of basic GUI agents' performance. Specifically, we train an **Intuitive Critic Model** (ICM) using positive and negative action examples from a base agent first. This critic evaluates the immediate correctness of the agent's intended actions, thereby selecting operations with higher success probability. Then, the initial critic guides agent actions to collect refined positive/negative samples, initiating the self-improving cycle. The augmented data then trains a second-round critic with enhanced discernment capability. We conduct experiments on various datasets and demonstrate that the proposed ICM can improve the test-time performance of various closed-source and open-source models, and the performance can be gradually improved as the data is recycled. The code and dataset will be publicly released.

## 1 Introduction

The automation of Graphical User Interface (GUI) interactions represents a critical frontier in developing intelligent digital assistants (Wang et al., 2024b; Hu et al., 2024; Nguyen et al., 2024). Recent breakthroughs in Large Vision-Language Models (LVLMs) (Wang et al., 2024a; Bai et al., 2025), leveraging advanced post-training techniques, have substantially enhanced agents' capabilities in interpreting natural language commands, perceiving visual elements, and executing multi-step tasks (Hong et al., 2024; Cheng et al., 2024). Within this rapidly evolving landscape, the development of robust GUI agents has largely converged on two primary methodological paradigms. The first approaches (Wu et al., 2025c; Xu et al., 2024; Qin et al., 2025; Liu et al., 2025a) train models through Supervised Fine-Tuning (SFT) to directly align their behavior with predefined task objectives. The second approaches employ Reinforcement Fine-Tuning (RFT) (Lu et al., 2025; Xia & Luo, 2025; Liu et al., 2025b), which significantly enhances generalization in complex tasks by adopting a reasoning format.

Despite these advances, the dynamic and continuous nature of real-world GUI tasks means that agents can still produce ambiguous or incorrect action proposals at any step. **A single mis-click or mis-typed output can be irreversible**, derailing the entire workflow and leaving the system in an unrecoverable state. This high-stakes environment imperatively demands **a mechanism for pre-execution validation.**

To avoid irreversible errors in execution and improve the performance of basic GUI agents during testing, previous studies have designed action verifiers for GUI agents (Wu et al., 2025b; Xiao et al., 2025; Yang et al., 2025), which are used to determine the correctness of multiple actions rolled out by GUI agents, and then filter out incorrect candidates. However, these existing implementations suffer from two primary limitations. First, training a correctness verifier requires defining positive and negative action samples. Existing work on defining negative samples relies on heuristic algorithms, such as randomly selecting click locations on the current screenshot (Xiao et al., 2025), which fails to capture the realistic action distribution and leads to suboptimal judgment performance. Second, the

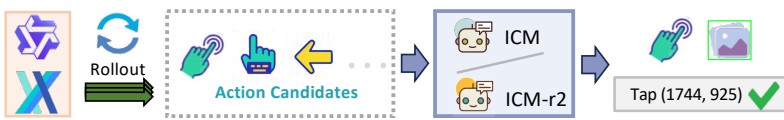

(a) The promotion process of the critic model to the GUI Agent during testing.

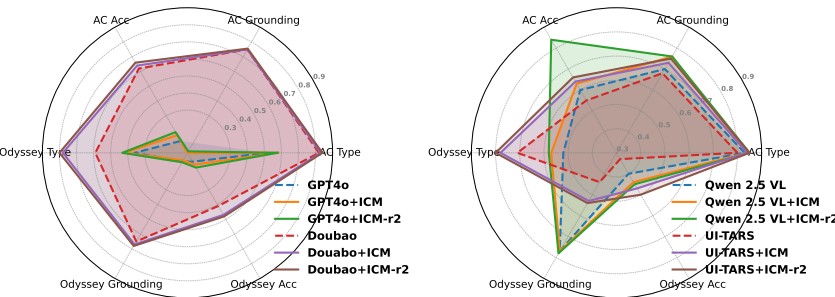

(b) Comparison of high-level task performance improvements on closed-source and open-source GUI Agents.

Figure 1: **Intuitive results.** ICM guides agents' action during testing, as shown in (a), thereby improving the agent's accuracy, and is continuously improved by the data flywheel, as shown in (b).

reasoning-based verifiers (Wanyan et al., 2025) implemented in existing work violate the intuitive properties of binary judgments. For an intuitive correctness judgment problem, biological research suggests that higher-level judgment pathways are often more adept than performing extensive multi-step reasoning (Liu & Pleskac, 2011; Poldrack et al., 2005; Doyon & Benali, 2005), which indicates that excessive reasoning can be less effective (Bilalić et al., 2008; Wan et al., 2011). Furthermore, reasoning-based judgment outputs more tokens, thereby reducing the efficiency of test-time scaling.

To fully leverage pre-execution evaluation to enhance GUI agent capabilities and execution correctness, we developed a **G**UI **A**ction Cr**i**tic's Dat**a** Flywheel System (GAIA). This system comprises two core phases: the initialization phase (Phase 1) and the iteration phase (Phase 2), yielding the **Intuitive Critic Model** (ICM). In **Phase 1**, we use real GUI agents to act on an existing dataset to collect positive and negative action data that are random but consistent with the behavior distribution. Using this binary-labeled action dataset, we train ICM to assess action correctness given environmental context. In **Phase 2**, as illustrated in Figure 1(a), ICM employs a Best-of-N approach to select the highest-probability correct actions from agent rollouts. While ICM guidance significantly improves action accuracy, challenging samples persist and produce errors. These difficult cases are annotated and fed back into the data flywheel. Through iterative data augmentation, the flywheel continuously incorporates new action samples, progressively covering challenging scenarios within the action space. Driven by this enriched dataset, we train an enhanced critic—Intuitive Critic Model on Round Two (ICM-r2)—which achieves higher discriminative accuracy for more precise behavioral guidance. This establishes a self-evolutionary virtuous cycle between the data flywheel and critic models, continuously improving GUI agent action accuracy.

Leveraging the proposed system GAIA, ICM achieves SOTA performance in action critique. Naturally, we integrate it into Test-Time Scaling (TTS) (Snell et al., 2025; Chen et al., 2024b; Snell et al., 2024; Prabhudesai et al., 2023; Wang et al., 2025; Tian et al., 2025) during inference, where ICM evaluates stochastically generated actions from the TTS process, releasing only executes the action if it is judged to be correct and has the highest probability of the correct token. Furthermore, based on GAIA's comprehensive definition, ICM can evaluate the correctness of actions across the entire space, rather than being limited to the accuracy of click actions in grounding tasks Yang et al. (2025); Wu et al. (2025a). To validate the framework's general applicability, we conduct joint experiments using mainstream GUI Agents (including GPT-4o (Hurst et al., 2024) and UI-TARS (Qin et al., 2025)) on several GUI Agent benchmarks.

As shown by the comparative results in Figure 1 (b), the guidance from our iteratively evolved critic models (ICM and ICM-r2) leads to significant performance improvements in basic GUI agents, including GUI operation task planning and grounding capabilities.

Figure 2: **Data flywheel curation pipeline for GAIA.** A sample dataset is constructed using GUI agent interactions. The positive and negative labels are marked by comparing the ground truth actions to train an action correctness discrimination model. After the critic model guides the GUI agent, it further expands the dataset, pushing the data flywheel to cover more action distributions, thereby promoting the iterative improvement of model performance.

Overall, the main contributions are summarized as follows:

1. We introduce GAIA—a novel Data Flywheel System designed for training GUI action-critic models. By iteratively curating positive and negative samples from real-world action data, GAIA continuously boosts model performance and robustness.

2. We propose the ICM for GUI interaction tasks, a critic model trained on data curated by our data flywheel. The ICM enhances the performance of existing GUI agents by employing a best-of-N approach to select the most probable correct action with TTS. This initial boost is then continuously refined as the ICM's discriminatory accuracy is iteratively improved by the data flywheel.

3. We comprehensively demonstrate across multiple datasets that ICM trained with our proposed GAIA system significantly enhances the overall performance of both closed-source and open-source GUI agents.

## 2 RELATED WORK

### 2.1 GUI AGENT

The development of autonomous agents powered by LLMs and LVLMs has significantly advanced interactive functionalities within digital environments. Early GUI systems primarily leveraged LLMs to interpret structured representations (Hong et al., 2024; Nong et al., 2024; Song et al., 2024). The development of LVLM simplifies the paradigm, allowing GUI agents to receive raw visual signals from the screenshots (Hu et al., 2024; Liu et al., 2024; Shen et al., 2024; Tang et al., 2025; Christianos et al., 2024; Zheng et al., 2025; Gou et al., 2024; Wu et al., 2025c). Recent efforts, such as Aguvis (Xu et al., 2024) and UI-TARS (Qin et al., 2025), have advanced autonomous GUI navigation by integrating explicit planning, sophisticated reasoning, and GUI-specific pretraining to handle complex digital environments. Concurrently, the advent of rule-based Reinforcement Learning (RL) approaches (Jaech et al., 2024; Guo et al., 2025) has further enhanced GUI agent capabilities. These RFT methods improve reasoning and generalization by enabling models to learn universal action strategies from high-quality samples (Liu et al., 2025c; Shen et al., 2025; Lu et al., 2025; Xia & Luo, 2025; Liu et al., 2025b). While fine-tuning and model scaling can enhance GUI agent capabilities, these methods are often prohibitively resource-intensive. This highlights a clear need for test-time enhancements that can offer universal performance improvements across various agent models without costly retraining.

## 2.2 CRITIC MODEL

To solve the problem of suboptimal single-shot model output (Zhang et al., 2025b; Martino et al., 2023; Wen et al., 2024; Chen et al., 2024a), research has gradually focused on improving the performance of the basic model during testing with the help of the critic model (McAleese et al., 2024; Ji et al., 2023; Kalyanpur et al., 2024; Zhang et al., 2025a; Xiong et al., 2025). This concept has been expanded to the GUI domain with notable works like GUI-Genie (Xiao et al., 2025), GUI-Actor (Wu et al., 2025b), GTA1 (Yang et al., 2025), and GUI-Critic-R1 (Wanyan et al., 2025). However, existing GUI critics often rely on synthetic data generated by heuristic algorithms, such as randomly selecting click locations (Wu et al., 2025b), cross-task substitution, or early truncation (Xiao et al., 2025). This approach fails to accurately simulate the complex behavior of real GUI agents across the full action space, thereby preventing the critic from learning faithful discrimination criteria. Furthermore, while some approaches use RL to inject reasoning capabilities into the critic (Wanyan et al., 2025), this often contradicts the very motivation for intuitive judgment (Liu & Pleskac, 2011; Wan et al., 2011) and introduces delays due to extended output token generation.

## 3 METHOD

In this section, we detail the design of our data flywheel-driven GAIA system for the GUI agent shown in Figure 2. We begin in Section 3.1 by introducing the general definition of the GUI agent task and the crucial role of the critic model. Section 3.2 delves into the design and application of our data flywheel system within the initial round of the evaluation process. In Section 3.3, we present the model training in the second round, which builds upon the outcomes from the first iteration and forms a virtuous cycle.

### 3.1 PRELIMINARIES

The interaction between a GUI agent and its environment can be formulated as a Markov Decision Process (MDP), denoted by the tuple $\langle \mathcal{S}, \mathcal{A}, \mathcal{Z}, \mathcal{T}, \mathcal{O} \rangle$. Here, $\mathcal{S}$ defines the state space of possible screen states, while $\mathcal{A}$ encompasses the action space, including interaction types like clicking, typing, and scrolling. The observation space $\mathcal{Z}$ captures inputs such as screenshots or structured UI representations. The state transition probability is given by $\mathcal{T} : \mathcal{S} \times \mathcal{A} \times \mathcal{S} \to [0, 1]$, mapping a state and action to a new state. Similarly, $\mathcal{O} : \mathcal{S} \times \mathcal{A} \to \mathcal{Z}$ describes the likelihood of observing a particular output given a state and an action. During GUI task execution, at each discrete time step $t$, the agent receives an input tuple $(z_t, u, h)$, comprising the current screen observation $z_t \in \mathcal{Z}$, the global task instruction $u$, and the accumulated interaction history $h$. The agent's decision-making process for GUI actions is then formalized by a structured policy function $\mathcal{F}$:

$$\mathcal{F}(z_t, u, h) \to o_t = \{a_t, c_t\}, \tag{1}$$

where $o_t$ represents the agent output at time $t$, consisting of the action type $a_t$ (e.g., click, scroll, and type) and its corresponding parameters $c_t$ (e.g., click coordinates, text content for typing). After $a_t$ is executed, the environment transitions to a new state $z_{t+1}$, and this iterative process continues until the task is successfully completed or a predefined termination condition is met.

The proposed ICM, building upon the same observations and the GUI agent's current proposed action $o_t$, outputs a judgment $j_t$ regarding the correctness of that action:

$$\mathcal{J}(o_t | (z_t, u, h)) \to \{j_t, p_t\}, \tag{2}$$

where $j_t$ is a binary indicator, "*correct*" for correct actions and "*wrong*" for incorrect, $p_t$ represents the probability of the judgment, which supports finding the correct action with the highest confidence. By enabling the sampling of multiple candidate actions and prioritizing them based on their respective correctness probabilities, ICM ensures that a more optimal action for the current state is selected and executed, significantly enhancing the agent's actual success rate.

### 3.2 ACTION DECISION WITH INTUITIVE CRITIC

#### 3.2.1 DATA CURATION

To enable the judgment model to distinguish the correctness of real actions, we meticulously define both positive and negative samples of GUI agent actions. We begin by having existing GUI agents

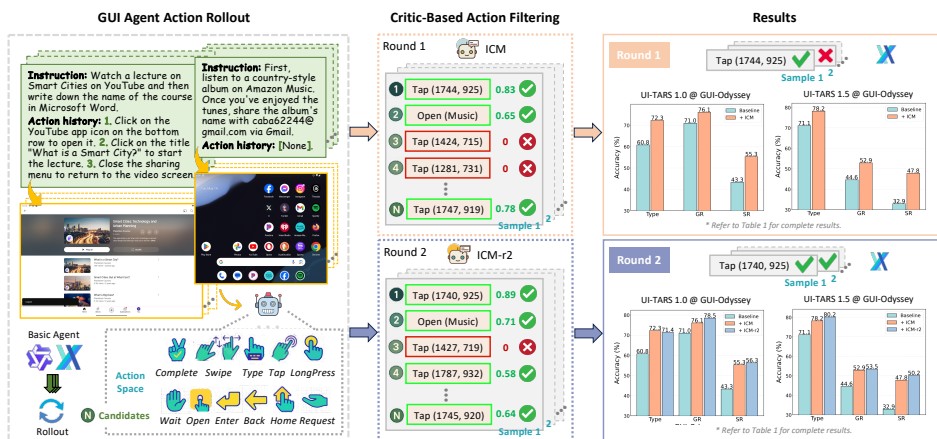

Figure 3: **Test-Time scaling pipeline.** Through best-of-N rollout, multi-candidate actions of GUI agents are given, and the correct action with the highest probability is selected after ICM evaluation.

$\Pi = \{\pi_1, \pi_2, ..., \pi_i\}$ (Qin et al., 2025) interact with and traverse publicly accessible datasets (Li et al., 2024; Lu et al., 2024), allowing us to collect authentic, step-level operations across various GUI scenarios. The datasets used are static and well-defined, meaning that each single-step action has a ground truth label to indicate the action the agent should take in the corresponding environment. Minor deviations in action parameters, such as slight offsets in click coordinates and semantic deviations in input text, are also guaranteed by recognized validation rules Wu et al. (2025c). For each action executed in a specific state $(z, u, h)$, we then leverage ground truth labels to determine its correctness. An action is designated positive (with a correctness judge $j =$ "*correct*") if it aligns with the GT.

Conversely, we identify negative samples (with a correctness score $j =$ "*wrong*") based on states where the agent's action deviates from the GT. This approach ensures that our collected negative operations are closely aligned with the actual error distribution observed in real GUI environments, significantly enhancing the quality and realism of our training dataset. To prevent bias during ICM training, we balance the collected positive and negative samples, ensuring an equal 50% split for each. This carefully curated dataset, denoted as $\mathcal{D} = \{j_k | z_k, u_k, h_k, o_k^{\pi_i}\}_{k=1}^{K}$, forms the foundation of our data flywheel GAIA.

### 3.2.2 ICM TRAINING AND GUIDANCE

Based on the dataset $\mathcal{D}$, ICM is trained to intuitively judge the correctness of actions. Specifically, the input of ICM includes the screen observation $z_k$, the instruction description $u_k$, the action history $h_k$, and the given agent action $o_k^{\pi_i}$. We implement ICM using LVLM and use standard cross-entropy loss to supervise the ICM's output tokens. For each sample in our dataset D, the model's output is a token representing either "*correct*" or "*wrong*". The training process aims to minimize the discrepancy between the model's predicted probability and the ground truth label:

$$\mathcal{L}_{\text{CE}} = -\frac{1}{K} \sum_{k=1}^{K} \Bigg[ j_k \log \left( P_{\theta_c}(\text{"}correct\text{"} \mid z_k, u_k, h_k, o_k^{\pi_i}) \right)$$

$$+ (1 - j_k) \log \left( 1 - P_{\theta_c}(\text{"}correct\text{"} \mid z_k, u_k, h_k, o_k^{\pi_i}) \right) \Bigg], \tag{3}$$

where $P_{\theta_c}(\text{"}correct\text{"} \mid z_k, u_k, h_k, o_k^{\pi_i})$ represents the probability assigned by the critic model $\theta_c$ to the "*correct*" token.

During test-time, a GUI agent $\pi_i$ generates $N$ candidate actions $\mathcal{O} = \{o_1, ..., o_N\}$ through N-rollout sampling. ICM evaluates these candidates by assigning each action a correctness judge $j_n$ and a potential confidence score represented by the token probability $p_n$. Leveraging the best-of-N filtering strategy, we select the optimal action $o^*$ from the subset of correct candidates $\mathcal{O}_{\text{correct}}$ that

Table 1: **GUI planning accuracy on AndroidControl and GUI-Odyssey.** $^{\dagger}$ represents the closed-source UI-TARS 1.5 called through the Doubao API. $^{*}$ represents an agent that reproduces the open-source model. ↑ and ↓ respectively represent the performance changes relative to the base agents.

| Model | Method | Size | AndroidControl-Low | | | AndroidControl-High | | | GUI-Odyssey | | |
|---|---|---|---|---|---|---|---|---|---|---|---|
| | | | Type | GR | SR | Type | GR | SR | Type | GR | SR |
| OS-Atlas-Base | ZS | 7B | 73.0 | 73.4 | 50.9 | 57.4 | 54.9 | 29.8 | 60.4 | 39.7 | 27.0 |
| SeeClick | SFT | 9.6B | 93.0 | 73.4 | 75.0 | 82.9 | 62.9 | 59.1 | 71.0 | 52.4 | 53.9 |
| Aria-UI | SFT | 3.9B | – | 87.7 | 67.3 | – | 43.2 | 10.2 | – | 86.8 | 36.5 |
| Aguvis | SFT | 7B | – | – | 80.5 | – | – | 61.5 | – | – | – |
| UI-R1 | RFT | 3B | 79.2 | 82.4 | 66.4 | 57.9 | 55.7 | 45.4 | 52.2 | 34.5 | 32.5 |
| GUI-R1 | RFT | 3B | 83.7 | 81.6 | 64.4 | 58.0 | 56.2 | 46.6 | 54.8 | 41.5 | 41.3 |
| GPT4o | | – | 78.8 | 8.0 | 20.4 | 52.4 | 3.6 | 13.2 | 36.6 | 11.6 | 11.4 |
| | + ICM | | 82.4 ↑3.6 | 9.2 ↑1.2 | 24.8 ↑4.4 | 58.0 ↑5.6 | 5.3 ↑1.7 | 17.0 ↑3.8 | 42.9 ↑6.3 | 10.0 ↓1.6 | 13.4 ↑2.0 |
| | + ICM-r2 | | 81.6 ↑2.8 | 8.5 ↑0.5 | 23.8 ↑3.4 | 57.6 ↑5.2 | 6.1 ↑2.5 | 18.8 ↑5.6 | 43.2 ↑6.6 | 11.4 ↓0.2 | 14.5 ↑3.1 |
| Doubao$^{\dagger}$ | | – | 97.0 | 86.4 | 86.2 | 82.0 | 75.0 | 62.4 | 67.1 | 67.3 | 43.8 |
| | + ICM | | 97.0 – | 86.6 ↑0.2 | 86.4 ↑0.2 | 83.2 ↑1.2 | 75.1 ↑0.1 | 64.4 ↑2.0 | 70.1 ↑3.0 | 68.4 ↑1.1 | 46.9 ↑3.1 |
| | + ICM-r2 | | 96.6 ↓0.4 | 86.6 ↑0.2 | 86.0 ↓0.2 | 84.2 ↑2.2 | 75.5 ↑0.5 | 66.0 ↑3.6 | 71.6 ↑4.5 | 68.0 ↑0.7 | 47.9 ↑4.1 |
| Qwen 2.5 VL | | 7B | 94.4 | 85.6 | 81.8 | 83.0 | 70.9 | 60.6 | 52.7 | 77.2 | 40.2 |
| | + ICM | | 95.4 ↑1.0 | 86.0 ↑0.4 | 81.2 ↓0.6 | 84.0 ↑1.0 | 75.9 ↑5.0 | 63.4 ↑2.8 | 57.3 ↑4.6 | 77.2 – | 43.7 ↑3.5 |
| | + ICM-r2 | | 94.6 ↑0.2 | 85.4 ↓0.2 | 81.8 – | 84.4 ↑1.4 | 75.9 ↑5.0 | 63.8 ↑3.2 | 58.1 ↑5.4 | 78.3 ↑1.1 | 44.8 ↑4.6 |
| UI-TARS 1.0$^{*}$ | | 7B | 90.0 | 85.1 | 75.4 | 80.8 | 68.9 | 58.2 | 60.8 | 71.0 | 43.3 |
| | + ICM | | 90.5 ↑0.5 | 87.6 ↑2.5 | 80.4 ↑5.0 | 82.3 ↑1.5 | 79.5 ↑10.6 | 67.5 ↑9.3 | 72.3 ↑11.5 | 76.1 ↑5.1 | 55.3 ↑12.0 |
| | + ICM-r2 | | 90.0 – | 86.8 ↑1.7 | 80.2 ↑4.8 | 82.7 ↑1.9 | 78.9 ↑10.0 | 67.1 ↑8.9 | 71.4 ↑10.6 | 78.5 ↑7.5 | 56.3 ↑13.0 |
| UI-TARS 1.5$^{*}$ | | 7B | 86.4 | 82.4 | 72.2 | 80.2 | 68.1 | 55.8 | 71.1 | 44.6 | 32.9 |
| | + ICM | | 90.3 ↑3.9 | 85.0 ↑2.6 | 79.0 ↑6.8 | 84.2 ↑4.0 | 73.3 ↑5.2 | 64.5 ↑8.7 | 78.2 ↑7.1 | 52.9 ↑8.3 | 47.8 ↑14.9 |
| | + ICM-r2 | | 90.1 ↑3.7 | 85.5 ↑3.1 | 79.2 ↑7.0 | 84.6 ↑4.5 | 74.7 ↑3.8 | 65.6 ↑7.4 | 80.2 ↑9.1 | 53.5 ↑8.9 | 50.2 ↑17.3 |

is judged as correct and whose corresponding *"correct"* token has the highest probability $p_n$:

$$o^* = \begin{cases} \arg \max_{o_n \in \mathcal{O}_{\text{correct}}} p_n, & \text{if } \mathcal{O}_{\text{correct}} \neq \varnothing \\ o_1. & \text{otherwise} \end{cases} \tag{4}$$

This approach effectively guides the agent to bypass single-shot output failures and select the most promising action, thereby significantly boosting its overall execution accuracy.

### 3.3 DATA FLYWHEEL AND CRITIC SCALING

Guided by Equation 4, the execution accuracy has been significantly improved. However, some difficult action samples require more precise judgment. Considering that ICM and test-time scaling performance can be further enhanced with data, we collect agent actions guided by ICM and, after filtering for positive and negative balance, add them to the data flywheel to form $\mathcal{D}^+ = \{j_k | z_k, u_k, h_k, o_k^{\pi_i}, \theta_c\}_{k=1}^{K'}$. $\mathcal{D}^+$ further covers the distribution of actions, providing a foundation for performance scaling.

Based on the challenging samples in this enriched dataset, we train the ICM on Round Two (ICM-r2), using the same cross-entropy loss as defined in Equation 3. This new dataset, which is specifically curated to expose the critic's most significant blind spots, allows ICM-r2 to acquire a more nuanced and accurate discriminative ability. Consequently, as illustrated in Figure 3, ICM-r2 provides more precise guidance for the agent's action selection, thereby fundamentally strengthening the critic's overall judgment and significantly improving the agent's performance on the most difficult tasks. Together with ICM, ICM-r2 demonstrates the power of a data flywheel-driven approach to stimulate the performance of GUI agents during testing.

## 4 EXPERIMENT

### 4.1 IMPLEMENTATION DETAILS

**Experimental Setup.** We use UI-TARS 1.0 (Qin et al., 2025) and UI-TARS 1.5 (Qin et al., 2025) for inference on the AndroidControl (Li et al., 2024) and GUI-Odyssey (Lu et al., 2024) training sets, and compare the real actions with GT to build $\mathcal{D}$ and $\mathcal{D}^+$.

Table 3: **GUI grounding accuracy on ScreenSpotV2.** $*$ indicates reproduced open-source agent performance. $\uparrow$ and $\downarrow$ respectively represent the performance changes relative to the base agents.

| Model | Method | Size | Mobile | | Desktop | | Web | | Avg. |
|---|---|---|---|---|---|---|---|---|---|
| | | | Text | Icon | Text | Icon | Text | Icon | |
| GPT-4o | ZS | – | 30.5 | 23.2 | 20.6 | 19.4 | 11.1 | 7.8 | 18.8 |
| OS-Atlas-Base | ZS | 7B | 93.0 | 72.9 | 91.8 | 62.9 | 90.9 | 74.3 | 82.5 |
| SeeClick | SFT | 9.6B | 78.0 | 52.0 | 72.2 | 30.0 | 55.7 | 32.5 | 53.4 |
| Aguvis | SFT | 7B | 95.6 | 77.7 | 93.8 | 67.1 | 88.3 | 75.2 | 84.4 |
| Qwen 2.5 VL | | 7B | 84.8 | 59.7 | 72.1 | 52.1 | 69.2 | 46.3 | 65.0 |
| | + ICM | | 87.9 $\uparrow 3.1$ | 70.1 $\uparrow 10.4$ | 79.4 $\uparrow 7.3$ | 57.1 $\uparrow 5.0$ | 74.7 $\uparrow 5.5$ | 49.2 $\uparrow 2.9$ | 70.4 $\uparrow 5.4$ |
| | + ICM-r2 | | 89.7 $\uparrow 4.9$ | 68.2 $\uparrow 8.5$ | 78.9 $\uparrow 6.8$ | 54.3 $\uparrow 2.2$ | 76.9 $\uparrow 7.7$ | 51.2 $\uparrow 4.9$ | 71.1 $\uparrow 6.1$ |
| UI-TARS 1.0$^{*}$ | | 7B | 93.1 | 82.4 | 94.8 | 76.4 | 91.8 | 84.2 | 88.1 |
| | + ICM | | 94.5 $\uparrow 1.3$ | 83.1 $\uparrow 0.7$ | 93.2 $\downarrow 1.6$ | 77.9 $\uparrow 1.5$ | 93.5 $\uparrow 2.0$ | 84.7 $\uparrow 0.5$ | 88.7 $\uparrow 0.6$ |
| | + ICM-r2 | | 94.2 $\uparrow 1.1$ | 83.7 $\uparrow 1.3$ | 95.7 $\uparrow 0.9$ | 78.7 $\uparrow 2.3$ | 92.1 $\uparrow 0.3$ | 84.7 $\uparrow 0.5$ | 89.0 $\uparrow 0.9$ |
| UI-TARS 1.5$^{*}$ | | 7B | 96.2 | 84.3 | 94.3 | 84.2 | 94.4 | 86.6 | 90.8 |
| | + ICM | | 96.5 $\uparrow 0.3$ | 85.3 $\uparrow 1.0$ | 95.4 $\uparrow 1.1$ | 84.3 $\uparrow 0.1$ | 94.2 $\downarrow 0.2$ | 85.2 $\downarrow 1.4$ | 90.2 $\downarrow 0.6$ |
| | + ICM-r2 | | 97.3 $\uparrow 1.1$ | 84.2 $\downarrow 0.1$ | 92.4 $\downarrow 1.9$ | 84.4 $\uparrow 0.2$ | 95.5 $\uparrow 1.1$ | 87.7 $\uparrow 1.1$ | 91.0 $\uparrow 0.2$ |

On the corresponding data, we develop the ICM and ICM-r2 based on Qwen2.5 VL 7B (Bai et al., 2025) and adopt the ms-swift (Zhao et al., 2024) framework for training. All action judgments followed the high-level approach, providing only global instructions to the ICM and ICM-r2, not single-step instructions. The distribution of the data flywheel is shown in Table 2. The critic model

Table 2: **Data distribution of the flywheel.** $\mathcal{D}$ and $\mathcal{D}^{+}$ respectively represent the data of the first and the second round of GAIA.

| Category | Source | Postive | Negtive |
|---|---|---|---|
| $\mathcal{D}$ | AndroidControl | 68.2k | 69.9k |
| | GUI-Odyssey | 65.4k | 66.8k |
| $\mathcal{D}^{+}$ | AndroidControl | (68.2+15.1)k | (69.9+14.0)k |
| | GUI-Odyssey | (65.4+26.1)k | (66.8+26.3)k |

guides the agents in the N-rollout process with $N = 8$. To allow the base agent to sample a reasonable range of potential actions, its temperature coefficient, top_k, and top_p are set to 1.0, 30, and 0.8, respectively. All experiments are conducted on 8 NVIDIA H100-80G GPUs.

**Evaluation.** To evaluate the performance of the agent after being guided by the critic model, we evaluate the agent's task understanding, grounding, and planning capabilities on the AndroidControl and GUI-Odyssey test sets. Furthermore, according to the input, the settings on AndroidControl can be divided into low-level tasks and high-level tasks. High-level tasks only input the global instruction to the agent, while low-level tasks will additionally input the single-step action plan. It should be noted that even for low-level agents, our critic model is guided only by high-level information to ensure the consistency of operation. GUI-Odyssey only adopts the high-level experimental setups. As for the agent's grounding ability, we measure and compare the performance on ScreenSpotV2 (Cheng et al., 2024).

**Comparison.** To verify the effectiveness of the proposed evaluation model in guiding existing agent models during testing, we selected a wide range of models. Among the closed-source models, we selected GPT4o (Hurst et al., 2024) and Doubao (UITARS 1.5) (ByteDance, 2025), and implemented action acquisition through API calls. For open source models, we reproduced Qwen 2.5 VL (Bai et al., 2025), UI-TARS 1.0 (Qin et al., 2025), and UI-TARS 1.5 (Qin et al., 2025), where Qwen 2.5 VL is a general multimodal understanding model and UI-TARS is a model fine-tuned for GUI agent tasks. We used the official prompt template to reproduce basic performance and test the superposition evaluation model. The original UI-TARS requires historical actions and up to 5 images of historical steps as input. To simplify the operation, we only let the model refer to the text description of the historical steps and discard the excessive historical image input. For detailed prompt and inference scripts, please refer to the appendix.

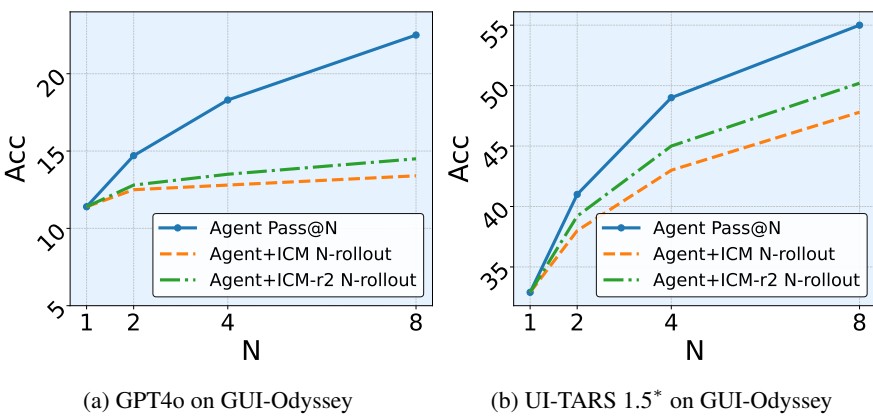

(a) GPT4o on GUI-Odyssey      (b) UI-TARS 1.5[*] on GUI-Odyssey

Figure 4: **Performance improvements of Pass@N and N-rollout.**

As a performance comparison, we selected Zero Shot (ZS) model OS-Atlas-Base (Wu et al., 2025c), SFT-tuned SeeClick (Cheng et al., 2024), Aria-UI (Yang et al., 2024), and Aguvis (Xu et al., 2024), and RFT-tuned UI-R1 (Lu et al., 2025) and GUI-R1 (Xia & Luo, 2025).

**Evaluation Metrics.** For planning tasks, in line with OS-Atlas (Wu et al., 2025c), we report action type prediction accuracy (Type), click point prediction accuracy (GR), and step success rate (SR). Specifically: **Type** measures the exact-match accuracy between predicted and ground-truth action types (e.g., "click" vs. "swipe"). **GR** evaluates grounding performance via click point prediction accuracy in specific action types (e.g., "click" and "long press"). **SR** is the step-wise success rate: a step is counted as successful only if both the predicted action and its associated arguments (e.g., click coordinates or input text) match the ground truth. For grounding tasks, we use click point prediction accuracy as our evaluation metric.

### 4.2 EXPERIMENTAL RESULTS

As shown in Table 1, the proposed ICM and ICM-r2 achieve extensive performance improvements for both zero-shot and fine-tuned GUI agents. On the AndroidControl-High test, ICM can improve agents' SR performance by up to $9.3\%$, while ICM-r2 can further improve it by an average of $1.32\%$. The same trend is also observed on GUI-Odyssey, demonstrating that existing models have the potential to correctly infer actions. ICM can leverage this potential to effectively improve existing agents during testing, and data flywheels can further amplify these improvements. For agents with lower basic performance, ICM can significantly improve their performance to an advanced level, which also demonstrates the potential of the model itself and the stimulation ability of the critic model. In terms of generalization, GPT4o, Doubao, and Qwen 2.5 VL were not included in the construction of the GAIA $\mathcal{D}$ and $\mathcal{D}^+$, but ICM and ICM-r2 still achieved significant performance improvements, demonstrating the inherent consistency of real action space data. The real action sampling in the proposed data flywheel effectively covers this space, providing effective support for critic training.

Both our ICM and ICM-r2 use a high-level approach to judge correctness and guide action, meaning the critic model is not aware of the current action plan. This setting is more consistent with practical applications, where only global instructions are given, and the agent must independently reason about each step's plan and action. For AndroidControl-Low, the agent is aware of the current action plan, resulting in higher baseline performance. Despite this, our ICM still achieves a certain degree of performance improvement, demonstrating the effectiveness of our proposed approach.

Table 3 shows the improvement of ICM and ICM-r2 on the grounding ability of agents on ScreenSpotv2. The ScreenSpotv2 data is not included in the proposed GAIA, and as a single-step operation, its environmental information is not completely consistent with the aforementioned datasets. Even so, our evaluation model still improves the performance of agents, which is sufficient to prove the validity of the data flywheel definition.

Please refer to the appendix for more visualization results.

Table 4: **Critic comparison.** The performance is calculated as (Qwen2.5 VL 7B w/ critic) - (Qwen2.5 VL 7B w/o critic).

| Model | N | AndroidControl-High | | |
| --- | --- | --- | --- | --- |
| | | $\Delta$ Type | $\Delta$ GR | $\Delta$ SR |
| UI-Genie-RM | 10 | – | – | 0.3 |
| ICM | 8 | 1.0 | 5.0 | 2.8 |
| ICM-r2 | 8 | 1.4 | 5.0 | 3.2 |

Table 5: **Impact of differences in critic model attributes on accuracy and guidance.**

| Model | Critic Acc | GUI-Odyssey | | |
| --- | --- | --- | --- | --- |
| | | Type | GR | SR |
| UI-TARS 1.5* | – | 71.1 | 44.6 | 32.9 |
| + RCM | 70.82% | 75.6 | 49.2 | 44.1 |
| + ICM | 83.19% | 78.2 | 52.9 | 47.8 |
| + ICM-r2 | 83.56% | 80.2 | 53.5 | 50.2 |

### 4.3 ABLATION STUDY

While utilizing ICM and ICM-r2, we used N-rollout to improve the test-time performance of existing GUI agents, where $N$ is set to 8 by default. To measure the impact of $N$ on final performance, we selected GPT4o and UI-TARS 1.5* as representative closed-source and open-source models, respectively, and compared their SR at different $N$ values on GUI-Odyssey. We also measured the models' Pass@N during the rollout process to reflect the model's performance ceiling. As shown in Figure 4, the increase in Pass@N accuracy reveals the potential of the agents themselves, while the evaluation model approaches this upper limit through N-rollout. The improvement in ICM-r2 and the gap between the upper limit provide potential performance gains for further cycles of GAIA.

### 4.4 QUALITATIVE EXPERIMENT

**Critic Model Comparison.** To evaluate the effectiveness of the proposed discriminant model, we compared the accuracy of the Qwen 2.5 VL using a best-of-N approach to guide inference on AndroidControl-High with UI-Genie-RM (Xiao et al., 2025). As shown in Table 4, due to the use of real action data, the proposed ICM significantly improves the accuracy of the base model, and ICM-r2 can further expand the advantage.

**Intuitive and Reasoning Critic.** To verify that the intuitive judgment proposed in this article is superior, this section implements a critic model based on reinforcement learning design. Specifically, the input of the Reasoning Critic Model (RCM) is consistent with ICM, and the output includes `<thinking>`...`</thinking>` and `<critic>`...`</critic>`, which are supervised by format reward and critic reward. The thought process emerges spontaneously from the model, and the critic reward represents the judgment on the correctness of the current action. The training of RCM is achieved through Group Relative Policy Optimization (GRPO). Considering the property of reinforcement learning, which is that it can stimulate model capabilities with less data, we randomly sampled 30k data from $\mathcal{D}^+$ to train RCM. This setting aligns with existing work on training critic models based on RL Wanyan et al. (2025), ensuring a fair comparison. This data includes samples from two rounds of GAIA and has the same distribution as the training data for ICM-r2. To intuitively compare the discriminative performance of different critic models, we collected the GAIA test set in a high-level manner on the AndroidControl and GUI-Odyssey test sets in the same way as we collected the training data.

Table 5 shows that the proposed ICM achieves an accuracy of 83.19% for correctness assessment, providing a foundation for action guidance. ICM-r2, benefiting from improved data quality, further achieves an accuracy of 83.56%. In contrast, RCM's classification accuracy is 70.82%, indicating that the thinking component fails to significantly contribute to the final assessment. In terms of action guidance accuracy, while UI-TARS 1.5* under RCM guidance outperforms the original model, it still falls short of ICM. This experimental result demonstrates that intuitive judgment outperforms reasoning for improving agents using a critic model. Besides, the RCM and GUI-Critic-R1 Wanyan et al. (2025) are required to generate Chain-of-Thought sequences enclosed in `<thinking>` tags before outputting a judgment, often consuming hundreds of tokens. In contrast, the intuitive-based ICM is trained to output a single token ("correct" or "wrong"). This disparity in output length results in an order-of-magnitude reduction in inference latency for ICM. For TTS, which requires evaluating multiple candidates, this efficiency is critical for practical deployment.

## 5    CONCLUSION

In this work, we addressed the critical challenge of high-stakes, irreversible errors in GUI agents by proposing a novel framework designed to unleash their latent potential at test time. Our GUI Action Critic's Data Flywheel System (GAIA) comprises a data flywheel that iteratively curates a dataset of realistic action samples and the Intuitive Critic Model (ICM) that evaluates action correctness. This framework establishes a self-evolutionary cycle: the flywheel continuously enriches its data, which in turn trains an increasingly powerful critic (ICM-r2). By leveraging a Best-of-N strategy, our ICM enables agents to select more reliable actions without the need for resource-intensive retraining. Experimental results on both closed-source and open-source agents demonstrate that GAIA provides significant performance gains in task planning and grounding capabilities, presenting a promising, scalable solution for building more robust and intelligent GUI agents.

In future work, we will consider unifying high-level and low-level guidance methods and collecting richer data in online testing, thereby continuously iterating the data flywheel and promoting the exploration of agent capabilities.

## 6    ETHICS STATEMENT

The research content of this paper is based on the LVLM GUI Agent. The research process of this paper does not violate ICLR ethics. There are no discrimination, bias, or fairness issues that need to be addressed. Our models are not expected to generate potentially harmful content.

## 7    REPRODUCIBILITY STATEMENT

This article studies a GUI Agent based on LVLM, focusing on proposing a critic model for existing agent actions. The base model and dataset used in this article are all from open-source and well-referenced, so this aspect does not affect the reproducibility. To further ensure reproducibility, we describe the parameters in detail in the main text Section 4.1 and the appendix, and provide prompts for all models involved in the appendix. We will release the source code and model checkpoints to support reproducibility.

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

## A  CLARIFICATION OF THE USAGE OF LLMs

This paper only used LLMs to assist and polish the writing. The retrieval, core innovation, method design, and experiments related to the paper were not conducted with the help of LLMs.

## B  INFERENCE PROMPTS

In this section, we introduce the inference parameters and prompt template of the LVLM used. The proposed GAIA data flywheel and trained ICM are used to guide existing GUI agents at test time. GAIA data is also constructed using the action reasoning of existing agents.

### B.1  GUI AGENT PLANNING TASK PROMPTS

**GPT4o** We use the API to test the closed-source model GPT4o (Hurst et al., 2024) as a GUI agent. Prompts refer to UI-TARS 1.0 (Qin et al., 2025) to ensure consistency in the action space. System prompts and user prompts refer to Prompt 1 and Prompt 3. The "You need to: {step_plan}" in User Prompt is only used when testing AndroidControl-Low (Li et al., 2024). In other conditions, the agent will not receive specific step instructions.

**Doubao**[†] Doubao, ByteDance's closed-source model interface, provides a closed-source version of UI-TARS 1.5, which represents the most advanced GUI agent. We test the performance of closed-source UI-TARS 1.5 by calling Doubao's API (ByteDance, 2025). The Prompt used for the test is consistent with the open source version UI-TARS 1.5 (Qin et al., 2025), see Prompt 2 and Prompt 3. The "You need to: {step_plan}" in User Prompt is only used when testing AndroidControl-Low. In other conditions, the Agent will not receive specific step instructions.

**UI-TARS 1.0**[*] For UI-TARS 1.0 (Qin et al., 2025), we use the open source weight UI-TARS-7B-DPO for testing. System prompts and user prompts refer to Prompt 1 and Prompt 3. The "You need to: {step_plan}" in User Prompt is only used when testing AndroidControl-Low. In other conditions, the agent will not receive specific step instructions. It should be noted that UI-TARS can receive up to five historical images as input. To simplify the process, we only use the text of "action_history" to describe the historical steps. For the image, we only input the current screenshot.

**UI-TARS 1.5**[*] For UI-TARS 1.5 (Qin et al., 2025), we use the open source weight UI-TARS-1.5-7B for testing. System prompts and user prompts refer to Prompt 2 and Prompt 3. The "You need to: {step_plan}" in User Prompt is only used when testing AndroidControl-Low. In other conditions, the agent will not receive specific step instructions. It should be noted that UI-TARS 1.5 can receive up to five historical images as input. To simplify the process, we only use the text of "action_history" to describe the historical steps. For the image, we only input the current screenshot.

**Qwen 2.5 VL** For Qwen 2.5 VL (Bai et al., 2025), we use the open source weight Qwen-2.5-VL-7B-Instruct for testing. We refer to the official use case and use the function calls to test Qwen's GUI Agent capabilities. The prompt is shown in Prompt 4 and 5.

### B.2  GUI AGENT GROUNDING TASK PROMPTS

Grounding capabilities are tested on the ScreenSpotV2 (Cheng et al., 2024) dataset. Because Grounding only provides single-step instructions and screenshots, and operations only involve click locations, the Prompts in Grounding differ from those in planning. The prompts for UI-TARS 1.0[*] and UI-TARS 1.5[*] refer to Prompt 6 and Prompt 7 respectively. The Qwen 2.5 VL test also applies function calls and constrains its output format through JSON, where Prompt is shown in Prompt 4 and 5. Prompt is the same as planning task, but without "action_history".

### B.3  ICM PROMPTS

The ICM and ICM-r2 trained on GAIA follow the same prompt and make judgments across the full action space. Therefore, the same prompt is used for both Planning and Grounding, as shown in Prompt 8. The information of "global_instruction" and "action_history" is consistent with that obtained by the basic GUI agent. "actor_set" describes the current action. If the action is click or

long press, it is constructed as "Tap at [x, y]", where x and y are the absolute coordinates of the click position in the original image. If the action is swipe, "actor_set" is constructed as "Swipe to up/down/left/right". If the action is type or open, "actor_set" is constructed as "Type/Open [text]", where[text] is the input text or the name of the App to be opened. The other actions have no parameters, so "actor_set" is directly the action name, such as "Wait", "Home", "Back", etc.

For the input image, we refer to the Set-of-Mark (SoM) approach (Yang et al., 2023). If the action is click or long press, a red circle is drawn at the click location. Otherwise, the original image is used directly as the ICM reference. The model's attention is implemented using FlashAttention (Dao et al., 2022). The data type is bfloat16. The epoch is 1, and the batch size is 16.

## C  TRAINING PARAMETERS

Using GAIA data, we train ICM and ICM-r2 with the following parameters. We fine-tune Qwen 2.5 VL 7B by inserting LoRA (Hu et al., 2022) into all linear layers, with lora_rank set to 8 and lora_alpha set to 32. The epoch is 1, and the batch size is 16. The optimizer is AdamW with a learning rate of 1e-4 and a warmup ratio of 0.05.

## D  BEST-OF-N METHOD

ICM and ICM-r2 use the Best-of-N approach to select the correct action with the highest probability from the N actions in the GUI agent rollout as the actual output, where the probability is expressed as the probability of the "correct" token. The core code of this process is shown in Code 1.

## E  VISUALIZATION RESULTS

We show the actions of the basic GUI agents on the sample, as well as the actions after being guided by ICM and ICM-r2. The comparison is shown in Figure 5 to 8.

**GPT4o and UI-TARS 1.0* System GUI Prompt**

```
You are a GUI agent. You are given a task and your
action history, with screenshots. You need to perform
the next action to complete the task.

## Output Format
Thought: ...
Action: ...

## Action Space

click(point='(x1 y1)')
long_press(point='(x1 y1)')
type(content='')
scroll(point='(x1 y1)', direction='down or up or right or left')
open_app(app_name=\'\')
drag(start_point='(x1 y1)', end_point='(x2 y2)')
press_home()
press_back()
finished(content='xxx')

## Note
- Use English in Thought part.
- Summarize your next action (with its target element) in one
sentence in Thought part.

## User Instruction
```

Prompt 1: GPT4o and UI-TARS 1.0* System GUI Prompt

**Doubao[†] and UI-TARS 1.5[*] System GUI Prompt**

```
You are a GUI agent. You are given a task and your
action history, with screenshots. You need to perform
the next action to complete the task.

## Output Format
Thought: ...
Action: ...

## Action Space

click(point='<|box_start|>(x1 y1)<|box_end|>')
long_press(point='<|box_start|>(x1 y1)<|box_end|>')
type(content='')
scroll(point='<|box_start|>(x1 y1)<|box_end|>',
    direction='down or up or right or left')
open_app(app_name=\'\')
drag(start_point='<|box_start|>(x1 y1)<|box_end|>',
    end_point='<|box_start|>(x2 y2)<|box_end|>')
press_home()
press_back()
finished(content='xxx')
```

```
## Note
- Use English in Thought part.
- Summarize your next action (with its target element) in one
sentence in Thought part.

## User Instruction
```

Prompt 2: Doubao[†] and UI-TARS 1.5[*] System GUI Prompt

**GPT4o, Doubao[†], UI-TARS 1.0[*] and UI-TARS 1.5[*] User GUI Prompt**

```
- User Instruction
{global_instruction}  (You need to: {step_plan})

- Action History
{action_history}

- Current Screenshot
{image_path}
```

Prompt 3: GPT4o, Doubao[†], UI-TARS 1.0[*] and UI-TARS 1.5[*] User GUI Prompt

**Qwen 2.5 VL GUI and Grounding Prompt**

```
You are a GUI Agent.

# Tools

You may call one or more functions to assist with the user query.

You are provided with function signatures within <tools></tools>
XML tags:
<tools>
{"type": "function",
 "function":
   {
   "name": "mobile_use", "description": "Use a touchscreen to
   interact with a mobile device, and take screenshots.
   * This is an interface to a
   mobile device with touchscreen. You can perform actions like
   clicking, typing, swiping, etc.
   * Some applications may take time to start or
   process actions, so you may need to wait and take successive
   screenshots to see the results of your actions.
   * The screen\'s resolution is 1092x2408.
   * Make sure to click any buttons, links, icons, etc with the

   cursor tip in the center of the element. Don't click boxes
   on their edges unless asked.",
  "parameters": {
  "properties": {
   "action": {
   "description": "The action to perform. The available
   actions are:
    * 'key': Perform a key event on the mobile device.
    - This supports adb\'s 'keyevent' syntax.
    - Examples: \\"volume_up\\", \\"volume_down\\", \\"power\\",
    \\"camera\\", \\"clear\\".
    * 'click': Click the point on the screen with
    coordinate (x, y).
    * 'long_press': Press the point on the screen with coordinate
    (x, y) for specified seconds.
    * 'swipe': Swipe from the starting point with coordinate
    (x, y) to the end point with coordinates2 (x2, y2).
    * 'type': Input the specified text into the activated
    input box.
    * 'system_button': Press the system button.
    * 'open': Open an app on the device.
    * 'wait': Wait specified seconds for the change to happen.
    * 'terminate': Terminate the current task and report its

    completion status.",
    "enum": ["key", "click", "long_press", "swipe", "type",
    "system_button", "open", "wait", "terminate"],
    "type": "string
    },
```

Prompt 4: Qwen 2.5 VL GUI and Grounding Prompt

**Qwen 2.5 VL GUI and Grounding Prompt (cont.)**

```
    "coordinate": {"description": "(x, y): The x (pixels from
      the left edge) and y (pixels from the top edge)
      coordinates to move the mouse to. Required only by
      `action=click`, `action=long_press`, and `action=swipe`.",
       "type": "array"},
    "coordinate2": {"description": "(x, y): The x (pixels from
      the left edge) and y (pixels from the top edge)
      coordinates to move the mouse to. Required only by
      `action=swipe`.", "type": "array"},
    "text": {"description": "Required only by `action=key`,
      `action=type`, and `action=open`.", "type": "string"},
    "time": {"description": "The seconds to wait. Required only
      by `action=long_press` and `action=wait`.",
      "type": "number"},
    "button": {"description": "Back means returning to the
      previous interface, Home means returning to the desktop,
      Menu means opening the application background menu,
      and Enter means pressing the enter. Required only
      by `action=system_button`",
    "enum": ["Back", "Home", "Menu", "Enter"], "type": "string"},
    "status": {
     "description": "The status of the task. Required only

        by `action=terminate`.",
     "type": "string", "enum": ["success", "failure"]
     }
    },
    "required": ["action"], "type": "object"
  }
  }
}
</tools>

For each function call, return a json object with function name
and arguments within <tool_call></tool_call> XML tags:
<tool_call>
{"name": <function-name>, "arguments": <args-json-object>}
</tool_call>

The user query:
{global_instruction}
Task progress (You have done the following operation on
the current device):
{action_history}

{image_path}
```

Prompt 5: Qwen 2.5 VL GUI and Grounding Prompt (cont.)

**UI-TARS 1.0* GUI Grounding Prompt**

```
You are a GUI agent. You are given a task and your action
history, with screenshots. You need to perform the next
action to complete the task.

## Output Format

Action: ...

## Action Space
click(point='<point>x1 y1</point>')

## User Instruction
{instruction}

{image_path}
```

Prompt 6: UI-TARS 1.0* GUI Grounding Prompt

**UI-TARS 1.5* GUI Grounding Prompt**

```
You are a GUI agent. You are given a task and your action
history, with screenshots. You need to perform the next
action to complete the task.

## Output Format

Action: ...

## Action Space
click(point='<|box_start|>(x1,y1)<|box_end|>')

## User Instruction
{instruction}

{image_path}
```

Prompt 7: UI-TARS 1.5* GUI Grounding Prompt

**ICM and ICM-r2 Critic Prompt**

```
You are an expert in evaluating the performance of a phone
operating agent. The agent is designed to help a user to
complete a task or retrieve information from the phone.
Given the user's task instruction, current action and current
screenshot, your goal is to decide whether the agent's current
action is correct or not.
Each action in the sequence is preceded by a corresponding
screenshot that captures the context in which the action occurs.

## Evaluation Criteria
Whether the agent's current action is correct and corresponding
to the user's task instruction.

## IMPORTANT
1. An action always follows a corresponding screenshot (even if
only the last few are provided).
2. If the current action is a tap on the screen, the point where
the action is clicked is marked with a red circle on
the screenshot.
3. You should whether answer [correct] or [wrong].

## Input
```

```
The input is given next, including global_task_instruction,
action_history, current_action, and screenshot.
The goal of the task (instruction): {global_instruction}
Action (plan) history: {action_history}
Current action of the agent: {actor_set}
Screenshot: {som_image_path}
```

Prompt 8: ICM and ICM-r2 Critic Prompt

**Best-of-N Example Code**

```python
# 1. construct critic input
texts = [
    critic_processor.apply_chat_template(msg, tokenize=False,
    add_generation_prompt=True) for msg in messages
]
image_inputs, video_inputs = process_vision_info(messages)
inputs = critic_processor(
    text=texts,
    images=image_inputs,
    videos=video_inputs,
    padding=True,
    return_tensors="pt",
)
inputs = inputs.to(critic_device)

# 2. generate output
output = critic_model.generate(**inputs,
                               do_sample=False,
                               max_new_tokens=2048,
                               return_dict_in_generate=True,
                               output_scores=True)
generated_ids = output.sequences

# 3. get token score
scores = output.scores[0]
critic_scores = scores[:,-2]

# 4. get output text: correct|wrong
for in_ids, out_ids in zip(inputs.input_ids, generated_ids):
    generated_ids_trimmed = [out_ids[len(in_ids) :]]
responses = critic_processor.batch_decode(
    generated_ids_trimmed, skip_special_tokens=True,
    clean_up_tokenization_spaces=False
)

# 5. find the index of the best action
max_score = -float('inf')
best_idx = -1
for idx in range(len(critic_outputs)):
    if responses[idx] == 'correct':
        if critic_scores[idx] > max_score:
            max_score = critic_scores[idx]
            best_idx = idx
```

Code 1: Best-of-N Example Code

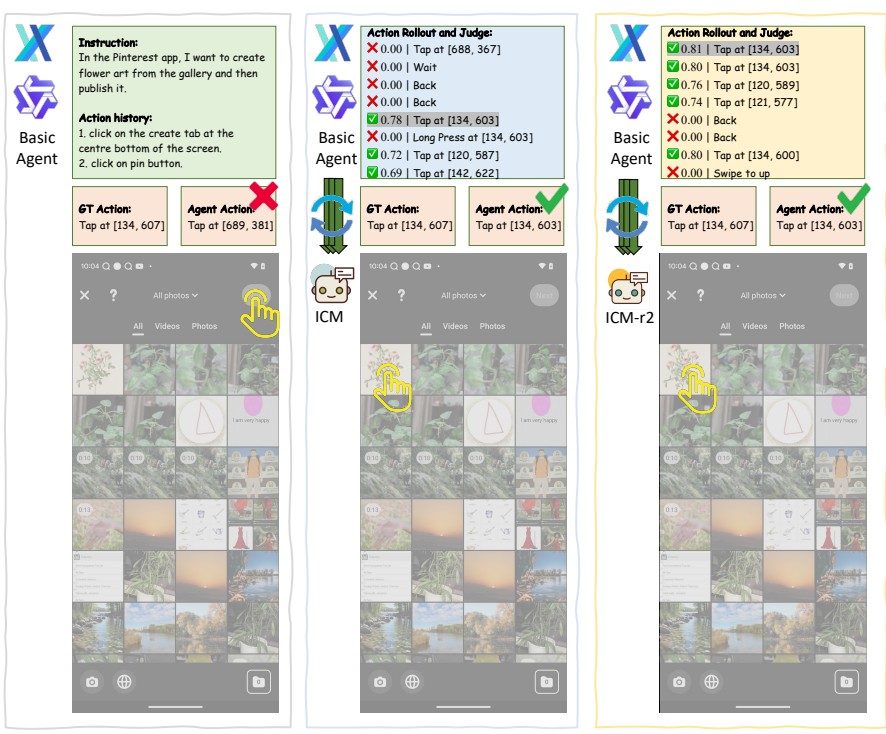

Figure 5: **Visualization result.** The basic agent selects the wrong action. Based on the action rollout, both ICM and ICM-r2 select the correct action from the candidates.

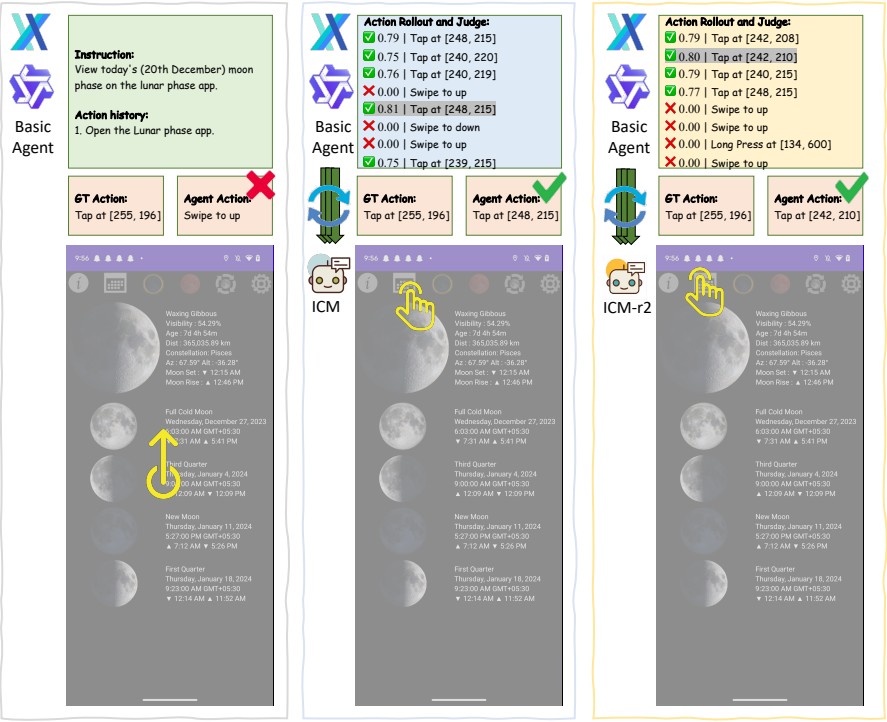

Figure 6: **Visualization result.** The basic agent selects the wrong action. Based on the action rollout, both ICM and ICM-r2 select the correct action from the candidates.

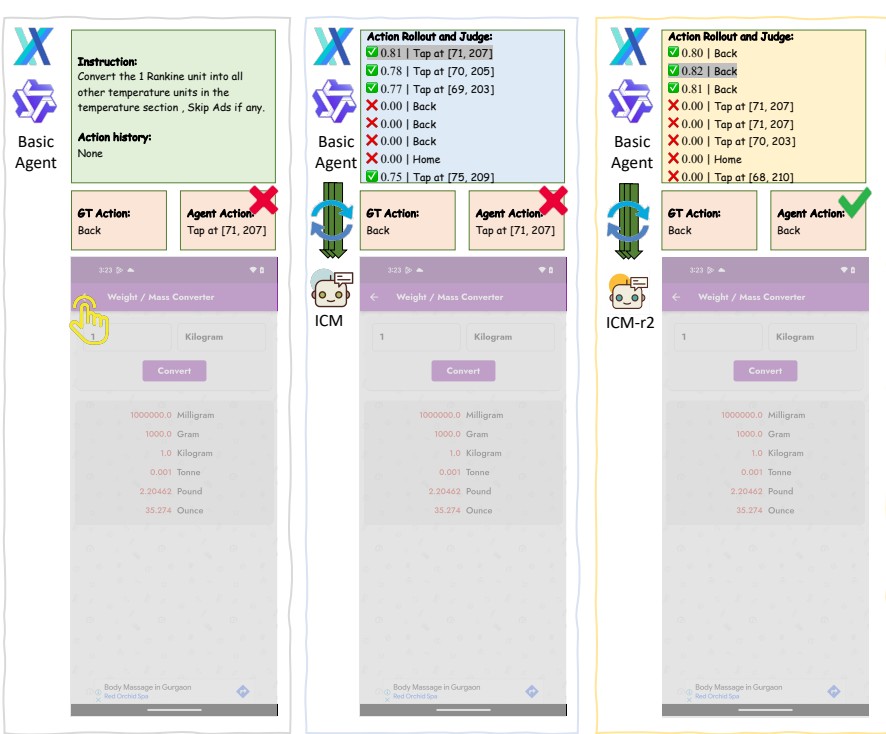

Figure 7: **Visualization result.** The basic agent selects the wrong action. ICM fails to select the correct one from the rollout candidates, while the enhanced ICM-r2 guides the correct selection.

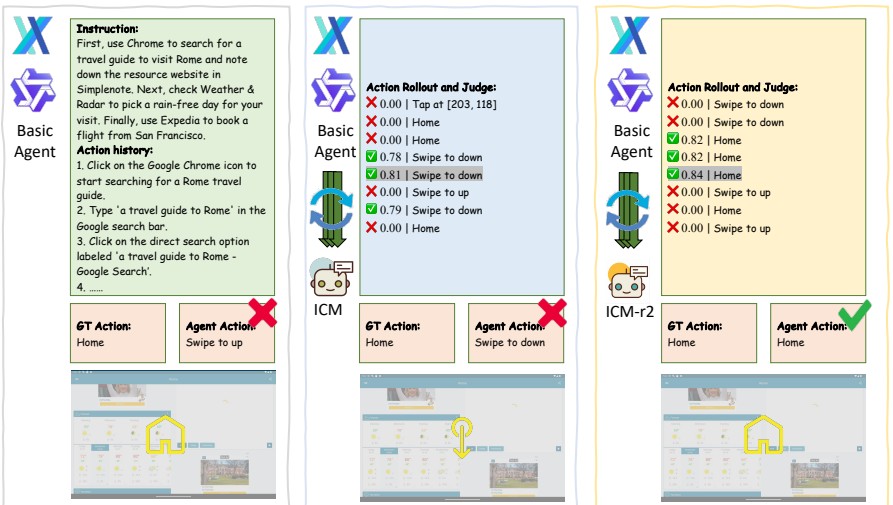

Figure 8: **Visualization result.** The basic agent selects the wrong action. ICM fails to select the correct one from the rollout candidates, while the enhanced ICM-r2 guides the correct selection.

