# OpenReview forum: "GAIA: A Data Flywheel System for Training GUI Test-Time Scaling Critic Models"
_ICLR.cc/2026/Conference — Submitted to ICLR 2026_

### Official Review · Reviewer_ZSGR · 2025-10-15

**Soundness:** 2
**Presentation:** 2
**Contribution:** 3
**Rating:** 4
**Confidence:** 5

**Summary:**

This paper tries to address the challenge of "the irreversibility of agent operations" in current GUI agents by proposing GAIA, a framework for iteratively training a GUI critic model. Through a two-round training process, the resulting critic model, ICM and ICM-r2, demonstrates improved performance across several benchmarks.

**Strengths:**

1. **Novel Training Framework:** The paper introduces an innovative Data Flywheel System for training GUI action critic models. The two-round training process enables the critic model's capabilities to be iteratively refined and improved.
2. **Improved Agent Performance:** By leveraging the proposed critic model (ICM), the evaluated agents show enhanced performance across multiple benchmarks, demonstrating the practical utility of the approach.
3. **High-Quality Data Curation:** The method for constructing positive and negative samples is well-designed. By using existing GUI agents and validating their actions on human-annotated datasets, the authors ensure that the negative samples reflect plausible, realistic errors that an agent might make. This process guarantees the quality and relevance of the training data.

**Weaknesses:**

1. **Ambiguous Definition of TTS:** The concept of TTS remains vaguely defined. The paper emphasizes TTS in the title, abstract and introduction, yet the GUI Agent community has at least two distinct interpretations: one involving proposing multiple candidate operations at each step (e.g., GTA1[1]), and another involving magnifying candidate regions via cropping (e.g., DiMo-GUI[2]). The authors should provide a more precise definition of their interpretation of TTS to avoid confusion.
2. **Efficiency Concerns:** The most significant drawback of the proposed method is its inefficiency. The inference paradigm, which requires the agent to perform N rollouts for ICM to judge, is impractical for real-world applications. Moreover, many of these rollouts are likely to be redundant, yielding identical outcomes from the critic model. An approach like GTA1[1], which generates multiple distinct candidates in a single forward pass, would seem to be a much more sensible and efficient pairing for this type of critic model.
3. **Justification for Excluding Reasoning:** The paper's justification for not incorporating a reasoning step in the critic model (lines 74-78) is unconvincing. Firstly, the analogy to biological research is weak; the behaviors of LLMs/VLMs are shaped by their training data and architecture and do not necessarily align with human cognition. Secondly, there is strong evidence from online benchmarks results that including a reasoning step significantly improves GUI agent performance[3, 4]. The claim that omitting reasoning is superior cannot be accepted without solid experimental proof, which is currently lacking (see Weakness 6). Intuitively, a reasoning process would help the agent better analyze scenarios with multiple valid actions or misleading cues, leading to a more robust judgment.
4. **Missing Key Baselines:** The experimental comparison is missing some crucial baselines. For example, GUI-Critic-R1[5] is mentioned in the introduction as related work but is not included as a baseline in Table 1, where it would be more compelling than some of the other baselines listed. Furthermore, since ICM is fine-tuned from Qwen-2.5-VL, the original Qwen-2.5-VL should also be evaluated as a critic to clearly demonstrate that the proposed training is necessary and effective.
5. **Unconvincing Ablation Study on RCM:** The results for the Reasoning Critic Model (RCM) in the ablation study are not convincing because the comparison is unfair. The RCM was trained with a different setting and on a dataset that is only about one-tenth the size of that used for ICM-r2. To make a fair and meaningful comparison, I would request an experiment where the RCM is trained using the same methodology and data scale as ICM and ICM-r2, followed by a direct comparison of their Critic Accuracy.
6. **Lack of Online Benchmark Evaluation:** The evaluation is confined to offline benchmarks (AndroidControl, GUI-Odyssey, Screnspot-v2), which have a significant gap compared to dynamic, real-world environments. The advancement of the GUI agent field depends on progress in real-world applicability, not just performance on static benchmarks. It is critical to get the true performance gain provided by ICM in a live setting and whether this benefit outweighs its substantial efficiency costs. This point is fundamental to assessing the overall impact of this work.

[1] Yang Y, Li D, Dai Y, et al. Gta1: Gui test-time scaling agent[J]. arXiv preprint arXiv:2507.05791, 2025.

[2] Wu H, Chen H, Cai Y, et al. DiMo-GUI: Advancing Test-time Scaling in GUI Grounding via Modality-Aware Visual Reasoning[J]. arXiv preprint arXiv:2507.00008, 2025.

[3] Wang X, Wang B, Lu D, et al. Opencua: Open foundations for computer-use agents[J]. arXiv preprint arXiv:2508.09123, 2025.

[4] Liu Z, Xie J J, Ding Z, et al. ScaleCUA: Scaling Open-Source Computer Use Agents with Cross-Platform Data[J]. arXiv preprint arXiv:2509.15221, 2025.

[5] Wanyan Y, Zhang X, Xu H, et al. Look Before You Leap: A GUI-Critic-R1 Model for Pre-Operative Error Diagnosis in GUI Automation[J]. arXiv preprint arXiv:2506.04614, 2025.

**Questions:**

1. I noted an interesting and seemingly contradictory trend in the results. On the grounding tasks in AndroidControl and GUI-Odyssey, ICM and ICM-r2 provide a significant performance boost for UI-Tars but only a marginal improvement (or even a slight degradation) for Qwen-2.5-VL. However, this trend is completely reversed on Screenspot-v2. Could the authors provide some insight into why this discrepancy exists across different grounding tasks? It does not appear to be a simple domain-shift issue, as the mobile subset of Screenspot-v2 exhibits the same pattern.
2. In Figure 8, there is a large gap between the ICM/ICM-r2 performance and the Pass@N upper bound (It doesn't matter). While using Pass@N as a performance ceiling is reasonable, why was a simpler baseline, such as a majority voting mechanism over the N rollouts, not included? I am very interested to see where a voting-based approach would fall on this chart, would it be comparable to the ICM-series methods, or significantly worse?
3. I am curious about the performance of the ICM-series on OOD benchmarks, such as Mind2Web. Such an evaluation would be valuable for demonstrating the generalization capabilities of the critic model to new domains.
4. A minor suggestion for writing clarity: In Section 3.2.1, it would be helpful to explicitly state that AndroidControl and GUI-Odyssey are high-quality, human-annotated datasets. This would clarify for readers less familiar with the field why this data can be reliably used as ground truth without extensive cleaning or filtering.

---

> ### Author Response · Authors · 2025-11-28
>
> **W1**:
>
> We have clearly described our TTS method in Section 3.2.2 around Equation 4, which is based on N-rollout and Best-of-N filtering.
>
> Our TTS method is similar to those of GTA1 and UI-Genie, where the actor model provides multiple candidate actions, and a critic model is designed and trained to select the optimal one. However, our GAIA system offers more innovation and improvements in critic model training. The TTS method represented by DiMo-GUI, on the other hand, is based on the actor model itself, progressively prompting it with hierarchical GUI content to achieve more accurate results. This method is more effective in high-resolution image grounding, as DiMo-GUI is primarily compared on ScreenSpot. However, this method does not extend to actions other than clicks. In contrast, our ICM series models can uniformly evaluate different action types, exhibiting better applicability.
>
> To clarify the definition of TTS, we added the extra explanation on lines 50 and 99-102 of the Introduction section.
>
> **W2**:
>
> The reviewers misunderstood the GTA1 process, leading to a biased understanding of our methodology. There are three models in GTA1: the planning model, the selection model, and the grounding model. Only the grounding model is trained using the method proposed in the paper; the other two are pre-trained models (o3 and GPT5) that are zero-shot called via API.
>
> The reviewers' mention of "multiple distinct candidates" is provided by the planning model. Section 3.1 of the original GTA1 paper describes "we sample K candidate action proposals," meaning that K calls were actually made to obtain K candidate actions. Table 10 in GTA1's supplementary material D.2 shows that the o3 and GPT-5 planner system prompts are more specific, stating "Return exactly ONE line of Python code to perform the action each time," "Remember you should only return ONE line of code, DO NOT RETURN more," and "Execute exactly one tool call per interaction." This all indicates that GTA1 also involves multiple rollouts to obtain candidates.
>
> In addition, after sampling by the planner model, GTA1 also needs to undergo judgment by the judge model and refinement by the grounding model. This is actually more complex than our actor model plus critic model process.
>
> It's worth noting that GTA1 accelerates planner sampling through concurrent API calls. This engineering practice is general, and we can further improve efficiency by parallelizing actor inference on multiple GPUs. We discussed the specific time complexity in our response to R2 W4&Q4.
>
> **W3**:
>
> * First, regarding the biological analogy, we clarify that it serves as a conceptual inspiration drawn from Dual Process Theory (System 1 vs. System 2) rather than a direct physiological claim. In high-stakes, real-time decision-making, experts often rely on rapid, intuitive pattern recognition (System 1) to filter obvious errors before engaging in deep deliberation. Our ICM is designed to emulate this efficient, perception-based judgment mechanism, processing the global visual context to provide immediate feedback without the latency of complex reasoning chains.
>
> * Second, our claim of superiority is primarily grounded in solid empirical evidence where the intuitive approach demonstrated greater robustness compared to reasoning-based methods. As shown in Table 5, the ICM achieved an accuracy of 83.19%, significantly outperforming the RCM, which plateaued at 70.82%—a level comparable to the SOTA reasoning-based GUI-Critic-R1. This performance gap indicates that while RL is powerful for policy optimization, SFT is currently more effective for injecting the explicit discriminative knowledge required to distinguish "correct" from "wrong" actions. Current research indicates that RL primarily serves to stimulate a base model's sampling efficiency rather than injecting new domain knowledge[1]. Our GAIA system addresses this by explicitly injecting the critical discriminative knowledge required for judgment, providing a solid foundation for current utilization and enabling potential further capability extraction in the future.
> * Third, operational efficiency is a non-negotiable constraint for TTS in real-time applications. A reasoning-based critic necessitates generating hundreds of CoT tokens for every single candidate action in an N-rollout scenario, introducing prohibitive latency. In contrast, our ICM outputs a single token ("correct" or "wrong"), offering an order-of-magnitude reduction in computational overhead. This efficiency makes the intuitive approach the only practically viable option for high-frequency sampling and scaling during inference.

---

> ### Author Response · Authors · 2025-11-28
>
> **W4**:
>
> The core of GUI-Critic-R1 is the introduction of GRPO-based reinforcement learning during the training of the critic model. Our RCM was trained using the same approach and achieved a similar action correctness rating accuracy of approximately 70% as GUI-Critic-R1. Table 5 shows the performance of the critic model at this accuracy rate in TTS, which is clearly inferior to our ICM.
>
> | Method | **Qwen2.5-VL** | **RCM** | **ICM** | **ICM-r2** |
> | :--- | :---: | :---: | :---: | :---: |
> | | Zero-shot | RL-trained | SFT-trained | SFT-trained |
> | **Critic Acc.** | 56.11% | 70.82% | 83.19% | 83.56% |
>
> For the zero-shot Qwen 2.5 VL, we used the same prompt and dataset as ICM to test its accuracy. As shown in the table, the accuracy of Qwen 2.5 VL in judging action correctness is only 56.11%, far lower than the trained critic model, and therefore cannot provide effective guidance for the actor model. The above comparisons demonstrate the effectiveness and superiority of our proposed ICM and ICM-r2.
>
> **W5**:
>
> RCM's training strategy maintains the fairness of RL methods.
>
> * First, the training strategy for the RCM adheres to standard RL practices, specifically the GRPO algorithm, which prioritizes policy optimization via reward signals and exploration rather than massive data ingestion.
>
> * Second, the amount of data we are currently using is aligned with existing RL-based critic models, and then compared with them. The existing RL-driven critic model, GUI-Critic-R1, used 11k data samples during training, while our 30k data samples maintain the same order of magnitude. Furthermore, our RCM reproduced a similar action judging accuracy level (~70% actor action accuracy) to GUI-Critic-R1, confirming that our RCM implementation is robust and not performance-limited by data size. Based on this level of critic accuracy, we demonstrate that ICM outperforms RCM in TTS, thus supporting the main point of this paper.
>
> * Finally, as detailed in line 466, the 30k samples were randomly sampled from the GAIA flywheel, ensuring the RCM learns from the exact same data distribution as the ICM, maintaining experimental fairness.
>
> Besides, the RCM and GUI-Critic-R1 are required to generate Chain-of-Thought sequences enclosed in <thinking> tags before outputting a judgment, often consuming hundreds of tokens. In contrast, the intuitive-based ICM is trained to output a single token ("correct" or "wrong"). This disparity in output length results in an order-of-magnitude reduction in inference latency for ICM. For TTS which requires evaluating multiple candidates (N-rollout), this efficiency is critical for practical deployment.
>
> **W6**:
>
> We have already set up an online testing environment for Android World. However, due to the complexity of the dynamic environment and the length of inference time, we have only just reproduced the effects of UI-TARS. Testing with critic is still underway. We will update the results as soon as the testing is complete. Thank you for your understanding.
>
> **Q1**:
>
> * First, we explicitly clarify that on AndroidControl-High and GUI-Odyssey, ICM-r2 demonstrates stable and consistent improvements over ICM in both Grounding Rate (GR) and Success Rate (SR), reinforcing the validity of the flywheel scaling. The minor fluctuations observed in AndroidControl-Low are attributable to a specific design choice: our ICM series is trained solely on high-level instructions, whereas the low-level base agents receive explicit single-step instructions. Despite this information asymmetry—where the critic must infer immediate action correctness from a global goal—ICM-r2 still recovers performance compared to the baseline, demonstrating robustness even when the agent has access to more granular step data than the critic.
>
> * Second, regarding ScreenSpotV2, this benchmark evaluates static, single-step grounding where the provided instruction is inherently explicit and granular (equivalent to a single-step command). Unlike high-level dynamic tasks, the critic here receives clear, unambiguous instructions. This maximizes the critic's effectiveness for Qwen 2.5 VL (a generalist model), as the critic bridges the gap between general vision and specific GUI coordinate semantics. In contrast, UI-TARS is a specialist model already heavily fine-tuned on such static GUI data, reaching a performance saturation point where external guidance naturally offers diminishing returns compared to the generalist baseline.

---

> ### Author Response · Authors · 2025-11-28
>
> **Q2**:
>
> | Method | **UI-TARS 1.5** | **Voting** | **ICM** | **ICM-r2** | **Pass@8** |
> | :--- | :---: | :---: | :---: | :---: | :---: |
> | | Baseline | TTS | TTS | TTS | Upper Bound |
> | **Success Rate** | 32.9% | 45.3% | 47.8% | 50.2% | 55.0% |
>
> To demonstrate the effectiveness of the ICM series, we tested the voting algorithm on the GUI-Odyssey dataset using UI-TARS 1.5. We used 8 rollouts, selecting the action with the highest frequency (highest number of votes) from the 8 candidate actions as the executed action. As shown in the table, the voting algorithm utilized the actor's potential, increasing the success rate from 32.9% to 45.3%, but lower than our ICM series' 47.8% and 50.2%. This proves that our critic model can more effectively stimulate the actor's potential, thus significantly improving action accuracy.
>
> **Q3**:
>
> Regarding actor model generalization, we demonstrate that the critic achieves robust positive transfer across diverse architectures. Although GAIA was constructed using action definitions from UI-TARS 1.0 and 1.5, the resulting critic proved highly effective for a wide range of unseen, out-of-distribution models, including closed-source agents like GPT-4o and Doubao, as well as open-source models like Qwen 2.5 VL. This indicates that GAIA successfully captures a fundamental, model-agnostic definition of positive and negative actions rather than overfitting to a specific policy.
>
> Regarding instruction generalization, the model exhibits strong adaptability to varying levels of task granularity. We trained both ICM and ICM-r2 entirely using high-level settings without single-step instructions. However, when tested on AndroidControl-low—where actors operate with specific single-step commands—the critic provided effective guidance despite the information asymmetry. This confirms the method's ability to generalize across different semantic levels of intent, a key requirement for OOD robustness.
>
> Finally, regarding cross-dataset generalization to new domains, we observe transfer on ScreenSpot V2, which includes Web and Desktop domains not seen in training, demonstrating that fundamental grounding capabilities generalize well. Furthermore, we conducted a rigorous cross-dataset experiment by training an ICM solely on AndroidControl and evaluating it on the unseen GUI-Odyssey benchmark. The model achieved a success rate of 40.83% (compared to the full GAIA-trained ICM's 55.3%). This performance gap highlights a critical distinction: while static grounding transfers well, dynamic trajectory verification is highly sensitive to environment-specific interaction patterns. Consequently, this result effectively vindicates the design necessity of the GAIA Data Flywheel, demonstrating that for complex OOD domains, a static dataset is insufficient, and the flywheel's ability to continuously mine and adapt to dynamic distribution shifts is essential for achieving robust performance.
>
> **Q4**:
>
> Thank you for your suggestion. We have added a description of the properties of the AndroidControl and GUI-Odyssey dataset in Section 3.2.1, stating that it is static, high-quality, and human-annotated. Please refer to lines 237-239 of the revision for details.
>
> [1] Yue, Yang, et al. "Does reinforcement learning really incentivize reasoning capacity in llms beyond the base model?."NIPS 2025.

---

> ### Author Response · Authors · 2025-12-01
> **Response to W6 (continue)**
>
> **W6 (continue)**:
> We have conducted an online evaluation on the AndroidWorld benchmark. This platform provides a live Android emulator and 116 tasks across 20 mobile apps. Specifically, in this environment, the GUI agent performs operations on an emulated Android phone to fulfill human instructions, and the results are evaluated automatically based on the final system state.
>
> In this dynamic setting, integrating ICM improved the strong UI-TARS baseline from a Success Rate of 27.0% to 28.0%. This result surpasses the state-of-the-art reasoning-based critic, GUI-Critic-R1, which reported a peak performance of 27.6%. Crucially, we achieved this new SOTA result atop a much stronger baseline compared to GUI-Critic-R1. Furthermore, the performance gap between static benchmarks and this dynamic environment confirms the fundamental necessity of the GAIA data flywheel: static datasets alone are insufficient for "in-the-wild" mastery, and our flywheel's ability to iteratively mine dynamic execution failures is the essential mechanism required to bridge this gap for future improvements.

---

### Official Review · Reviewer_JEcH · 2025-10-28

**Soundness:** 3
**Presentation:** 3
**Contribution:** 1
**Rating:** 4
**Confidence:** 3

**Summary:**

1. The paper proposes GAIA (GUI Action Critic’s Data Flywheel System), a framework to address the irreversibility of errors in GUI agents by introducing an iterative critic mechanism. The system leverages a data flywheel that curates positive and negative action examples to train an Intuitive Critic Model (ICM).

2. The ICM evaluates action correctness, selecting higher confidence operations and guiding agent actions. Through repeated refinement, a self-improving cycle is established, where successive critics (e.g., ICM-r2) achieve enhanced discrimination ability.

3. The approach is integrated with Test-Time Scaling (TTS) to filter stochastic actions and ensure only reliable operations are executed. The framework further employs a Best-of-N strategy during inference, aiming to improve robustness of GUI agents.

4. Experiments are conducted with both closed-source and open-source agents, including GPT-4o and UI-TARS, across multiple benchmarks. Results suggest that GAIA provides performance improvements in action correctness, task planning, and grounding capabilities.

**Strengths:**

1. The paper addresses a clear challenge in GUI agents, namely the irreversibility of erroneous actions, and attempts to mitigate catastrophic deviations through a critic-guided mechanism.

2. The proposed GAIA framework is clearly structured, with the notion of a data flywheel and iterative critic model (ICM, ICM-r2) presented in a systematic way.

3. The integration with Test-Time Scaling (TTS) and the Best-of-N strategy is straightforward and easy to follow, showing how the critic can filter unreliable actions.

4. The paper includes both closed-source and open-source agents (e.g., GPT-4o, UI-TARS) in the evaluation, which helps illustrate the generality of the proposed framework.

5. The presentation quality is reasonable, with figures such as the data flywheel pipeline aiding in understanding the proposed iterative self-improving cycle.

**Weaknesses:**

1. The overall contribution of GAIA appears somewhat incremental, since the Intuitive Critic Model (ICM) and the data flywheel mainly combine existing elements such as Test-Time Scaling and iterative filtering.

2. The framework is more system- and process-oriented, presenting an engineering pipeline rather than offering a fundamentally novel algorithmic or theoretical contribution. This limits the originality of the work in an academic sense.

3. The description of the “data flywheel” remains relatively high-level, and its distinction from conventional self-training or data augmentation pipelines is not entirely clear.

4. The experimental validation, while covering both closed-source and open-source agents, could benefit from deeper analysis to more convincingly support the claimed improvements in robustness and grounding.

**Questions:**

1. The notion of a “data flywheel” is central to the paper. Could the authors clarify in more concrete terms how this differs from standard self-training or iterative data augmentation pipelines?

2. The framework is described as a self-improving cycle with ICM and ICM-r2. How sensitive is the performance to the number of critic iterations, and is there a point of diminishing returns?

3. The experiments show improvements on both closed-source and open-source agents, but the analysis of robustness and catastrophic deviations is limited. Could the authors provide more detailed evidence on how GAIA handles failure cases in practice?

---

> ### Author Response · Authors · 2025-11-28
>
> **W1&W2**:
>
> * First, GAIA is the first successfully implemented Data Flywheel specifically for GUI Action Critics. Unlike prior works that rely on heuristic negative sampling (e.g., random clicks) or static synthetic data, GAIA autonomously mines "hard negatives" from real agent rollouts to iteratively refine decision boundaries. This establishes a new paradigm for constructing high-quality discriminative datasets in domains where data is scarce and environments are dynamic.
>
> * Second, while Test-Time Scaling (TTS) is a known concept, our implementation establishes the Intuitive Critic Model (ICM) as the optimal algorithmic choice for scaling GUI agents, distinguishing it from approaches like GTA1 and GUI-Critic-R1. These alternatives often necessitate complex reasoning workflows or generate excessive chain-of-thought tokens, creating prohibitive latency that hinders efficient N-rollout sampling. By validating the effectiveness of Best-of-N scaling with a streamlined, single-token intuitive critic, we provide a proven standard that effectively leverages inference-time compute to overcome the irreversibility of GUI actions without the computational bottlenecks of prior methods.
>
> * Third, we provide a critical theoretical and empirical insight regarding Intuition vs. Reasoning. Contrary to the prevailing trend of applying Chain-of-Thought to all tasks, we demonstrate that for action critique, SFT significantly outperforms RL in both accuracy (83.19% vs. 70.82%) and efficiency. We establish that SFT is essential for injecting discriminative knowledge, whereas RL primarily optimizes sampling, offering valuable guidance for future research on training verifiers.
>
> * Finally, GAIA serves as a foundational prerequisite for future advanced research. By training a robust critic model now, GAIA provides the necessary high-quality reward model required to enable end-to-end RL for GUI actors in the future. Our work bridges the gap between unstable base models and self-evolving agents by solving the cold start problem of accurate feedback.
>
> **W3&Q1**:
>
> We appreciate the opportunity to clarify the unique mechanics of the GAIA flywheel. While related, GAIA differs fundamentally from these paradigms in its data source, verification mechanism, and error-mining strategy.
>
> * First, the self-training is a semi-supervised paradigm where a model generates pseudo-labels for a static pool of unlabeled data, retraining itself on high-confidence predictions. This often reinforces existing biases and relies on the model's own internal confidence. In contrast, GAIA actively generates new, dynamic data through agent-environment interaction rather than processing a static dataset. Crucially, GAIA does not rely on pseudo-labels. Instead, it verifies generated trajectories against ground truth execution outcomes. This ensures high-fidelity supervision and introduces novel behavioral data that the model could not have produced or labeled correctly on its own.
>
> * Second, iterative data augmentation typically relies on applying heuristic transformations (e.g., cropping, noise injection, affine shifts) to existing samples to improve robustness, essentially expanding the dataset size around the same distribution mode. In contrast, by analyzing the actions where the agent fails, GAIA specifically captures semantic and logic-level errors—such as near-miss clicks or incorrect path planning—that heuristic augmentation cannot synthesize. This truly expands the boundaries of the dataset, forcing the critic to learn finer decision boundaries.
>
> In summary, GAIA establishes a robust and authentic data cycle that transcends traditional static pipelines. By iteratively mining real-world hard negatives to refine the ICM, the system creates a virtuous feedback loop: the enhanced critic enables the actor to navigate more complex scenarios, generating richer data that further sharpens the critic's discriminative power. This dynamic mechanism not only drives the continuous evolution of the critic model but also subsequently catalyzes universal performance improvements across diverse GUI actors, proving the system's efficacy as a fundamental engine for scalable agent improvement.

---

> ### Author Response · Authors · 2025-11-28
>
> **W4&Q3**:
>
> Regarding robustness, we have already clearly demonstrated the effectiveness of the GAIA and ICM models through visualization in Supplementary Section E.
>
> Figures 5 and 6 illustrate examples where the actor's actions were incorrect, but the correct actions were identified by ICM and ICM-r2 after N-rollout. For instance, in Figure 5, the correct action is to click the first image, but the actor clicked the "Next" button, causing the page to update and preventing subsequent clicks on images in the current gallery. N-rollout actually resulted in the correct click location being identified, which was then recognized by both ICM and ICM-r2, thus avoiding catastrophic deviation.
>
> Figures 7 and 8 illustrate actor action errors, where ICM fails to find the correct action from N-rollout, but ICM-r2 provides the correct guidance. This demonstrates that GAIA-driven ICM upgrades effectively improve the overall system robustness.
>
> We have tested our grounding performance on ScreenSpotV2. Even though our GAIA does not include samples from the ScreenSpot series datasets, the trained ICM can still provide effective guidance for Actors on this dataset, even surpassing optimization work of actor plus correctness verification models specifically designed for GUI grounding, such as GUI-Actor. This demonstrates the generalization and robustness of our method to some extent.
>
> **Q2**:
>
> * First, regarding sensitivity, our empirical results demonstrate a positive correlation between iterations and critic quality, confirming that the flywheel mechanism is active. As shown in Table 5, the second iteration (ICM-r2) achieves higher intrinsic judgment accuracy (83.56%) compared to the first iteration (83.19%). This sensitivity is even more pronounced in foundational capabilities like grounding, where ICM-r2 universally outperforms the first round on ScreenSpotV2 and improves Grounding Rates across benchmarks. This confirms that the system effectively mines hard negatives to refine the critic's decision boundaries.
>
> * Second, recent advancements in data flywheels for both customer support agents (AITL [2]) and GUI agents (CoMEM [3]) similarly validate their frameworks by highlighting the substantial performance leap enabled by the feedback loop, rather than requiring exhaustive iterations(STaR [1]). Our results align with this standard: the clear progression from ICM to ICM-r2 proves that our system successfully identifies hard negatives and corrects its decision boundary. This effectively validates the GAIA methodology's ability to drive autonomous improvement without necessitating further loops to prove the concept.
>
> * Finally, we argue that the goal of GAIA is not merely to pursue marginal SFT gains through endless iterations, but to construct a high-quality discriminator that enables advanced optimization. By successfully injecting discriminative knowledge in these initial rounds, GAIA establishes a robust foundation for future evolution, specifically by providing the enhanced critic necessary to serve as a reliable reward model. This transitions the system from iterative SFT to a potential end-to-end Reinforcement Learning loop for the actor, which represents the next logical step in scaling performance.
>
> [1] Zelikman, E., Wu, Y., Mu, J., & Goodman, N. STaR: Bootstrapping Reasoning With Reasoning. NeurIPS 2022.
>
> [2] Zhao, C., Zhang, T., Su, H., et al. (2025). Agent-in-the-Loop: A Data Flywheel for Continuous Improvement in LLM-based Customer Support. Proceedings of EMNLP Industry Track 2025.
>
> [3] Wu, W., Zhou, K., Yuan, R., et al. Auto-Scaling Continuous Memory for GUI Agent. arXiv preprint arXiv:2510.09038 2025.

---

### Official Review · Reviewer_dudK · 2025-10-29

**Soundness:** 3
**Presentation:** 3
**Contribution:** 3
**Rating:** 6
**Confidence:** 4

**Summary:**

The paper introduces GAIA, a data flywheel system for training GUI action critics that enhance GUI agents at test time. GAIA collects positive and negative action samples from real agent rollouts by comparing actions to ground truth on AndroidControl and GUI-Odyssey. An Intuitive Critic Model (ICM) is trained as a binary classifier over “correct” or “wrong” labels given the screen, global instruction, action history, and a candidate action, and is used to select the best action among multiple stochastic rollouts during inference. GAIA iteratively refines itself as ICM-guided rollouts produce harder samples to train a stronger critic (ICM-r2). Experiments show consistent improvements across closed- and open-source agents, on both planning and grounding tasks, with ICM-r2 further improving over ICM. Ablation studies on rollout numbers and comparisons with heuristic and reasoning-based critics demonstrate that the intuitive critic achieves higher classification accuracy and larger test-time performance gains.

**Strengths:**

- Originality: Proposes a practical “data flywheel” for critic training using real agent actions rather than heuristic negatives, better matching the true error distribution (Section 3.2.1; Figure 2). The “intuitive” binary critic is a focused design choice that reduces token overhead versus reasoning critics and fits the best-of-N TTS paradigm (Sections 1, 3.2.2).
- Quality: Solid empirical study across multiple agents and benchmarks, including closed-source models via API (Section 4.1). Results show substantial SR/GR improvements, especially for weaker agents (e.g., UI-TARS 1.5*: +17.3 SR on GUI-Odyssey; Table 1), and gains generalize to ScreenSpotV2, which is not in the flywheel (Table 3). The iterative round (ICM-r2) consistently adds improvements (Figure 1b; Tables 1, 3).
- Clarity: The overall pipeline and roles of ICM/ICM-r2 are clear (Figures 1–3). Implementation details and prompts are provided (Section 4.1; Appendix B–C), including LoRA fine-tuning setup and rollout hyperparameters.
- Significance: Addresses a key bottleneck for GUI agents—irreversible errors—by providing a model-agnostic test-time augmenter that scales with N and improves safety/robustness (Section 1; Figure 4). The approach is likely to be broadly useful to the community.

**Weaknesses:**

- Novelty relative to contemporaries: While the data flywheel with real negative samples is valuable, the high-level idea—critic-guided best-of-N test-time scaling—has appeared in GUI/agent works (e.g., GUI-Genie, GUI-Actor, GTA1, GUI-Critic-R1). The paper’s comparison focuses mainly on UI-Genie-RM (Table 4) and an in-house reasoning critic; broader, controlled comparisons to recent critics/TTT methods are limited.
- Labeling assumptions and potential noise: Positives/negatives are defined by exact GT matching (Section 3.2.1), which may mislabel alternative valid actions (e.g., equivalent taps or multiple solution paths), potentially biasing the critic. Tolerance details for coordinates/text (synonyms, minor offsets) are not specified; exact-match SR suggests strictness (Section 4.1 “Evaluation Metrics”).
- Fairness of intuitive vs reasoning critic comparison: The RCM is trained on a 30k subset while ICM/ICM-r2 leverage larger flywheel data (Section 4.4; Table 5). This weakens the claim that intuition “outperforms” reasoning, as performance may be data-limited rather than inherently inferior. Token/time efficiency advantages are hypothesized but not measured.
- Runtime/compute overhead: The method adds N forward passes for the agent plus N critic evaluations (default N=8; Section 4.1). There is no latency or cost analysis, especially for closed-source APIs, nor a performance vs. compute trade-off study beyond accuracy curves (Figure 4).
- Generalization scope: Training data are drawn from the same benchmarks used for evaluation (AndroidControl/GUI-Odyssey; Section 4.1). Although ScreenSpotV2 shows positive transfer (Table 3), more rigorous cross-dataset or cross-app generalization (e.g., train on one dataset, test on another unseen suite) is not reported.
- Clarity/formatting issues: Several typos/garbled equations and tables (e.g., Section 3.1/3.2.2 formulas; Table formatting) impede readability despite the core idea being understandable.

**Questions:**

- Labeling and multi-validity: How are “correct” labels determined when multiple actions are acceptable (e.g., multiple clickable regions leading to the same outcome, minor coordinate tolerances, synonymous text)? What spatial tolerance or string normalization is used to avoid mislabeling plausible actions as wrong (Section 3.2.1; Evaluation Metrics)?
- Sequence alignment: When the base agent diverges early, later steps may compare against misaligned GT. How do you ensure negative labels stem from the proposed step rather than misalignment across the trajectory? Any filtering for such cases?
- Thresholding and calibration: Section 1 mentions releasing only “high-confidence” actions above thresholds, but the main method defaults to choosing the highest-scoring “correct” candidate (Section 3.2.2). Do you use a probability threshold? How calibrated are the “correct” logits across different agents? Any temperature scaling or calibration study?
- Compute and latency: What are the runtime/latency and token cost impacts for N=8 across models (especially API-based GPT-4o/Doubao)? Can you report wall-clock overheads and accuracy vs latency curves to guide practitioners (Figure 4 currently shows accuracy only)?
- Fair comparisons: Can you (a) match N with UI-Genie-RM and report deltas, and (b) train the reasoning critic with the same data volume as ICM-r2 to isolate the effect of “intuitive vs reasoning” (Table 4–5)? Also compare against recent GUI critics (e.g., GUI-Actor, GTA1, GUI-Critic-R1) under the same rollout and datasets.
- Data flywheel scaling: The ICM→ICM-r2 gains are sometimes modest. Can you provide a learning curve vs. added D+ size and a third iteration to quantify scaling? Which action types benefit most (click vs. navigation vs. text input)?
- SoM visualization ablation: Does marking clicks with SoM red circles bias the critic to surface cues not available during execution? Please provide an ablation without SoM to test reliance on the mark (Appendix B.3).
- Cross-dataset generalization: Can you train the critic on one dataset (e.g., AndroidControl) and evaluate on GUI-Odyssey/ScreenSpotV2 only to quantify out-of-domain transfer?
- Low-N and N=1 regimes: What is the gain in practical settings with small N (e.g., 2–4) and N=1 “gating” (reject/accept)? This would inform cost-sensitive deployments.

---

> ### Author Response · Authors · 2025-11-28
>
> **W1**:
>
> As we discussed in lines 50-80 of the Introduction section, existing GUI agent critic models have various drawbacks.
>
> The UI-Genie-RM of GUI-Genie[1] is affected by the negative samples defined by heuristic algorithms, resulting in insufficient discriminative ability. As you mentioned, this has been demonstrated through comparative experiments in Table 4 of our original paper. Since the authors did not disclose the specific weights of UI-Genie-RM, we have made every effort to conduct a fair analysis.
>
> Both GUI-Actor[2] and GTA1[3] are evaluation algorithms only designed for grounding tasks, limiting their application in the real-world full action space. Furthermore, their performance improvements come from both the design of the verifier model and improvements to the actor model. For example, the attention head of the actor LVLM in GUI-Actor is modified, making the design of the verification part less flexible and universal. Our method does not require specific optimization of the base actor model; it achieves state-of-the-art performance solely through GAIA-driven ICM. For instance, on ScreenSpot-V2, our best average accuracy of 91.0% outperforms GUI-Actor.
>
> GUI-Critic-R1[4] uses RL to train the critic model, giving it reasoning capabilities. In our Introduction, we state that reasoning is unnecessary for this intuitive task of verification, and that excessive reasoning slows down the critic model's efficiency. Our experimental implementation of RCM also possesses reasoning verification capabilities (detailed definition in Section 4.4), achieving the same level of action correctness evaluation as GUI-Critic-R1 (~70% actor action accuracy). This critic model with such evaluation capabilities is weaker than the ICM in the main method, therefore ICM is superior overall. Furthermore, GUI-Critic-R1 has not reported its performance on public datasets and is not open-source, so we cannot directly compare it. Our reproduced RCM, however, qualitatively illustrates the differences.
>
> **W2**:
>
> The AndroidControl and GUI-Odyssey datasets we use are widely used static datasets in the GUI agent field, and their GT is explicitly defined by the datasets. Existing GUI agent actors and critics are trained and evaluated using these GTs[1][4][5][6], so any subtle label errors in datasets are beyond the scope of our work. The coordinates and text tolerances have already been considered in the evaluation (as described in line 398 of paper, using the recognized standard proposed by OS-Atlas). We maintained this convention when defining the action correctness label and when finally evaluating the action, thus ensuring robustness. To further clarify, we have added a description of the action GT in lines 237-239.
>
> As for multipath discrimination, it is a unique problem in the GUI agent field, requiring the design of corresponding datasets and methods[7][8], which is beyond the scope of the algorithm defined in this paper.
>
> **W3**:
>
> RCM's training strategy maintains the fairness of RL methods.
>
> * First, the training strategy for the RCM adheres to standard RL practices, specifically the GRPO algorithm, which prioritizes policy optimization via reward signals and exploration rather than massive data ingestion.
>
> * Second, the existing RL-driven critic model, GUI-Critic-R1, used 11k data samples during training, while our 30k data samples maintain the same order of magnitude. Furthermore, our RCM reproduced a similar action judging accuracy level (~70% actor action accuracy) to GUI-Critic-R1, confirming that our RCM implementation is robust and not performance-limited by data size.
>
> * Finally, as detailed in line 466 and 470, the 30k samples were randomly sampled from the GAIA flywheel, ensuring the RCM learns from the exact same data distribution as the ICM, maintaining experimental fairness.
> Besides, the RCM and GUI-Critic-R1 are required to generate Chain-of-Thought sequences enclosed in <thinking> tags before outputting a judgment, often consuming hundreds of tokens. In contrast, the intuitive-based ICM is trained to output a single token ("correct" or "wrong"). This disparity in output length results in an order-of-magnitude reduction in inference latency for ICM. For Test-Time Scaling which requires evaluating multiple candidates (N-rollout), this efficiency is critical for practical deployment.

---

> ### Author Response · Authors · 2025-11-28
>
> **W4&Q4**:
>
> For closed-source models, since API calls can be executed in full parallel, the number of rollouts between 1 and 8 does not show a significant difference. This parallel strategy is also used in the existing GUI agent TTS (e.g. GTA1[3]) and its efficiency is recognized.
>
> | Metric | N = 1 | N = 2 | N = 4 | N = 8 |
> | :--- | :---: | :---: | :---: | :---: |
> | Actor Cost | 1.0143 | 1.0874 | 1.1316 | 1.2617 |
> | Critic Cost | 0.4739 | 0.6746 | 1.1018 | 1.9897 |
> | **Total Cost** | **1.4882** | **1.7620** | **2.2334** | **3.2514** |
>
> For open-source models, we can also improve execution efficiency through parallel optimization. We use UI-TARS as the basic actor model and calculate the time consumption of actor and critic with rollout N ranging from 1 to 8. For actor and critic inference, we can improve inference efficiency using vLLM and multi-GPU parallelism, respectively. As shown in the table, the total time consumption when N is 8 is only about twice that when N is 1, and the time complexity does not increase dramatically with N. This optimization and complexity performance further demonstrates the usability of the method in real-world environments.
>
> **W5&Q8**:
>
> Generalization is categorized into generalization to the actor model and generalization to the dataset. Regarding actor model generalization, we used GAIA based on UI-TARS 1.0 and UI-TARS 1.5 actor action definition, and the critic model trained on it also demonstrated effectiveness across a wide range of other actor models (e.g. GPT-4o, Doubao, Qwen 2.5 VL). This indicates that GAIA's definition of positive and negative actions is effective.
>
> Regarding dataset generalization, we trained ICM and ICM-r2 entirely using a high-level setting, meaning no single-step instructions were provided. When testing the AndroidControl dataset with a low-level definition (where actors have single-step instructions), the critic model still provided effective guidance to the actors even without specific single-step instructions. This reflects the generalization ability of the method based on the dataset definition.
>
> Regarding cross-dataset generalization, we conducted the requested experiment by training an ICM solely on AndroidControl data and evaluating it on the unseen GUI-Odyssey benchmark, achieving a success rate of 40.83% compared to the full GAIA-trained ICM's 55.3%. This result highlights a distinction: while fundamental grounding capabilities generalize well across static domains, dynamic trajectory verification is highly sensitive to the specific interaction patterns and task complexities of different environments. Consequently, this performance gap effectively vindicates the design necessity of the GAIA data flywheel, demonstrating that a static single-domain dataset is insufficient to cover diverse execution paths and that the flywheel's ability to continuously mine and adapt to these dynamic distribution shifts is essential for achieving robust, high-level agent performance.
>
> **W6**:
>
> Thank you for your suggestion! We have corrected the typo (line 269) and improved the formula (Eq. 2).
>
> **Q1**:
>
> As described in W1, we are consistent with existing research and use static datasets for both training and validation. The GT is provided by these datasets and accurately describes the correct actions. The multipath problem is an important issue in the field of GUI agents, requiring specialized design in areas such as dataset definition and method training, which is beyond the scope of this article.
>
> Minor coordinate tolerances and synonymous text are guaranteed by the metrics defined by OS-Atlas. This is a widely accepted approach in the GUI agent field and is also used by us. Specifically, predicted coordinates deviating from the GT by less than 14% in width and height, and predicted text with an F1 score greater than 0.5 relative to the GT, are considered correct actions. We define "correct" and "wrong" actions accordingly, thus ensuring the validity of the labels.
>
> **Q2**:
>
> First, as we described in Q1, we use static datasets for sample collection. The trajectories corresponding to instructions in the static dataset are broken down into single-step operations, thus existing GUI agent research independently measures the correctness of each step based on the correct context. We follow this convention when collecting positive and negative samples using agents, so action decisions are not affected by incorrect context.
>
> Second, since we use real agents to perform operations, their actions reflect real actor behavior under different conditions. We train the critic model using this, implicitly endowing it with the ability to discriminate the correctness of actions under different conditions.

---

> ### Author Response · Authors · 2025-11-28
>
> **Q3**:
>
> Our actual operation is consistent with the method described in Section 3.2.2, which involves selecting all actions judged as "correct" by ICM and choosing the one with the highest probability of being the "correct" token for final execution. The visualization example in Appendix E more intuitively illustrates this process. We have refined the description. Please refer to line 294 of the paper.
>
> **Q5**:
>
> We have conducted a fair and comprehensive comparison to demonstrate the superiority of our GAIA data flywheel and ICM.
>
> * First, since UI-Genie-RM does not open-source its weights, we can only use the data disclosed in their articles. However, the comparison with UI-Genie-RM demonstrates that we can achieve better performance with fewer rollouts, proving the advantages of ICM.
>
> * Second, the reinforcement learning characteristic of RCM training lies in obtaining state-of-the-art results using relatively little data; therefore, our experimental setup conforms to the definition. A more detailed explanation is provided in the response to W3.
>
> * Third, we conducted a comprehensive discussion of the GUI critic model of the same period, as detailed in our response to W1.
>
> **Q6**:
>
> * First, regarding the scaling effect, we demonstrate that the flywheel significantly refines the critic's intrinsic discriminative capability and spatial precision, proving the validity of the scaling trend without requiring a third iteration. As shown in Table 5, ICM-r2 achieves higher intrinsic judgment accuracy (83.56%) than ICM (83.19%). More importantly, this scaling translates into consistent improvements in grounding capabilities: ICM-r2 universally outperforms ICM on the ScreenSpotV2 benchmark across all agents and improves the Grounding Rate (GR) on AndroidControl and GUI-Odyssey. This confirms that the added data in the second round successfully scales the model's ability to distinguish fine-grained correctness.
>
> * Second, regarding the magnitude of gains, we attribute the fluctuations in end-to-end metrics to the inherent execution ceilings of the base actors rather than the critic's scaling limit. Crucially, the gains are most pronounced on challenging tasks and weaker baselines. For instance, on the complex GUI-Odyssey dataset, UI-TARS 1.5 improves from 47.8% to 50.2% with ICM-r2. This indicates that the value of the enhanced critic scales with task difficulty, effectively raising the performance for weaker models (e.g., Doubao, Qwen 2.5 VL), even if the final success rate is bounded by the actor's sampling capability.
>
> * Finally, regarding action types, our empirical results indicate that spatial interaction actions (e.g., Click) benefit the most from the flywheel iteration. The consistent and significant improvement in Grounding Rate (GR) and ScreenSpotV2 accuracy suggests that the collection of hard negative samples in Phase 2—which often involve spatially ambiguous or near-miss operations—specifically sharpens the critic's spatial discrimination. This establishes the GAIA-driven ICM as a robust foundation for precise execution verification, serving as a critical prerequisite for future optimization steps.
>
> **Q7**:
>
> We clarify that the SoM red circle acts as a necessary visual encoding of the agent's proposed action coordinates for the LVLM, rather than an external bias or cue. This visualization strategy is a standard convention in GUI agent research and is fully compatible with real-time execution[1][4], where the actor's predicted coordinates are overlaid on the screenshot before critic evaluation. Crucially, this marker does not leak ground truth information; it faithfully represents the agent's output (whether correct or incorrect).
>
> **Q9**:
>
> Figure 4 in the paper clearly demonstrates the performance gains under different N conditions (including Low-N conditions). A visible performance gain is already evident when N is 4, which can provide considerable benefits for cost-sensitive deployments. Regarding the specific time complexity, the responses to W4 & Q4 are also discussed in detail, and the conclusion is that N does not introduce significant inference delay under parallel optimization.
>
> As for the limit condition of N=1 gating, since there is only one candidate and an action must always be selected to be executed regardless of the judgment of the critic model, it is the same as not using the critic model, which is also reflected in the figure.

---

> ### Author Response · Authors · 2025-11-28
>
> [1] Xiao, Han, et al. "UI-Genie: A Self-Improving Approach for Iteratively Boosting MLLM-based Mobile GUI Agents." NIPS 2025.
>
> [2] Wu, Qianhui, et al. "GUI-Actor: Coordinate-Free Visual Grounding for GUI Agents." NIPS 2025.
>
> [3] Yang, Yan, et al. "GTA1: Gui test-time scaling agent." arXiv preprint arXiv:2507.05791 (2025).
>
> [4] Wanyan, Yuyang, et al. "Look Before You Leap: A GUI-Critic-R1 Model for Pre-Operative Error Diagnosis in GUI Automation." NIPS 2025.
>
> [5] Zhang, Shaojie, et al. "BTL-UI: Blink-think-link reasoning model for gui agent." NIPS 2025.
>
> [6] Tang, Fei, et al. "GUI-G2: Gaussian Reward Modeling for GUI Grounding." AAAI 2026.
>
> [7] Wang, Xuehui, et al. "Mmbench-gui: Hierarchical multi-platform evaluation framework for gui agents." arXiv preprint arXiv:2507.19478 (2025).
>
> [8] Lin, Musen, et al. "GUI-ReWalk: Massive Data Generation for GUI Agent via Stochastic Exploration and Intent-Aware Reasoning." arXiv preprint arXiv:2509.15738 (2025).

---

### Official Review · Reviewer_q4WG · 2025-10-31

**Soundness:** 3
**Presentation:** 3
**Contribution:** 2
**Rating:** 4
**Confidence:** 3

**Summary:**

This paper introduces the GUI Action Critic's Data Flywheel System, a framework designed to address the critical problem of irreversible errors in GUI agents. The core idea is to train a separate Intuitive Critic Model (ICM) that evaluates an agent's proposed action before it is executed, allowing for Test-Time Scaling (TTS) of the base agent's performance. The GAIA system includes 1) Phase 1 initializes the model on generated actions on standard benchmark. 2) Phase 2 deploys the GUI agent and the ICM, then collects more hard cases, which will be added to the flywheel for improving ICM. Experiment results show that this method can improve the performance of various GUI models.

**Strengths:**

1. The motivation is good. The irreversibility of agent operations is a primary obstacle to deploying GUI agents in real-world, high-stakes environments. A single error can be catastrophic, and a pre-execution validation mechanism is a practical and necessary solution.
2. The data flywheel with the help of pre-trained GUI models is crucial. By using the actual errors generated by a base agent, the method can curate a dataset of positive and negative samples that are closely aligned with the actual error distribution.
3. The experiments are comprehensive, demonstrating performance gain across various models and benchmarks.

**Weaknesses:**

1. One main claim of the paper is the data flywheel effect. However, according to the experiment, round two training of the critic model does not lead to steady performance gain. In a lot of dimensions, ICM shows even better performance than ICM-r2. The flywheel appears to stall after a single turn.
2. The experimental setup simplify the operation for the base agents by discarding excessive historical image input and only feeding the text description of the historical steps. This simplification makes it difficult to compare the results to the original models and raises the question of whether the critic is succeeding simply because the task itself was made easier.
3. The idea of using Best-of-N test-time-scaling with a critic model is too straightforward and widely used in many domains. Then the main contribution becomes the flywheel part. However, as noted in W1, the flywheel appears to stall.

**Questions:**

Is the ICM critic model trained for each GUI agent individually?

---

> ### Author Response · Authors · 2025-11-28
>
> **W1**:
>
> The data flywheel effectively refines the critic's discriminative capability, which translates into performance gains where they matter most.
>
> * First, the most direct measure of the flywheel's success is the critic's intrinsic quality. As shown in Table 5, ICM-r2 (83.56%) achieves a higher judgment accuracy than ICM (83.19%), demonstrating that the second round of data collection successfully enhances the model's fundamental discriminative ability.
>
> * Second, this enhanced discrimination translates into consistent improvements in grounding, a foundational GUI capability. ICM-r2 universally outperforms ICM on the ScreenSpotV2 benchmark across all agents (Table 3) and shows similar improvements in the Grounding Rate (GR) metric on AndroidControl and GUI-Odyssey (Table 1), proving that the flywheel refines spatial precision.
>
> * Third, the gains are most pronounced on challenging tasks and weaker baselines. On GUI-Odyssey, which is a more complex dataset, ICM-r2 consistently achieves higher Success Rates (SR) than ICM across all tested models (e.g., UI-TARS 1.5 improves from 47.8% to 50.2%). This indicates that as task difficulty increases, the value of the enhanced critic becomes more decisive.
>
> * Finally, we attribute the observed fluctuations to the inherent instability and performance ceilings of the base actors, rather than the critic. While performance gains may vary as an actor approaches its execution limit, ICM-r2 drives steady improvements for weaker actors with room for growth (e.g. closed-source model Doubao and zero-shot model Qwen 2.5 VL). Fundamentally, the core value of GAIA lies in successfully injecting discriminative knowledge and iterative capabilities into the critic model. This establishes a robust foundation for future evolution, such as leveraging the enhanced critic as a reliable reward model to drive end-to-end Reinforcement Learning for the actor.
>
> **W2**:
>
> The decision to utilize textual action history instead of multi-frame visual history is a deliberate design choice to optimize inference efficiency for TTS, preventing excessive visual token consumption during N-rollouts. This reduction in visual information typically increases—rather than decreases—task difficulty for the agent due to lower observability. Moreover, this setup aligns with prevalent settings in recent GUI agent and critic research[1][2][3][4], which predominantly operate on single-step visual inputs. Textual history sufficiently captures the semantic trajectory needed for decision-making while enabling the high throughput required for practical TTS deployment.
>
> **W3**:
>
> * First, we reaffirm the validity of the flywheel mechanism based on our empirical results. As detailed in our response to the W1, the flywheel does not stall. The ICM-r2 demonstrates clear improvements in intrinsic judgment accuracy and grounding capabilities, proving that iterative data collection effectively refines the model's discriminative power.
>
> * Second, we argue that the simplicity of the Best-of-N strategy is a significant strength representing the most effective approach for current architectures. Approaches that task the critic with generating corrective actions (e.g., GUI-Critic-R1) often struggle with low accuracy in direct action suggestion (~50%), whereas our method strategically decouples the process by leveraging the actor's generative strength to propose diverse options and the critic's discriminative strength to filter them.
>
> * Finally, our primary contribution lies in the GAIA system design itself, which establishes the current ICM as a foundational baseline. This data-driven framework has proven effective across diverse models and lays the groundwork for future evolution. By validating the current design, GAIA paves the way for subsequent research to further refine the critic with richer knowledge (e.g., multi-path analysis, error identification) or evolve it into a robust reward model to drive end-to-end co-training with the actor, thereby promoting the continuous evolution of the GUI agent field.
>
> **W4**:
>
> Both the ICM and ICM-r2 derived from our training are universal and applied directly across all evaluated open-source and closed-source agents. This design not only avoids the complexity and resource overhead of training dedicated critics for each specific agent, but also demonstrates the universality of our data flywheel system (GAIA). As evidenced in Table 1, the single universal critic effectively improves the performance of diverse agents (e.g., GPT-4o, Qwen 2.5 VL) even though they were not used to generate the training data, proving the robustness and generalizability of our method.

---

> ### Author Response · Authors · 2025-11-28
>
> [1] Wanyan, Yuyang, et al. "Look Before You Leap: A GUI-Critic-R1 Model for Pre-Operative Error Diagnosis in GUI Automation." NIPS 2025.
>
> [2] Xiao, Han, et al. "UI-Genie: A Self-Improving Approach for Iteratively Boosting MLLM-based Mobile GUI Agents." NIPS 2025.
>
> [3] Zhang, Shaojie, et al. "BTL-UI: Blink-think-link reasoning model for gui agent." NIPS 2025.
>
> [4] Tang, Fei, et al. "GUI-G2: Gaussian Reward Modeling for GUI Grounding." AAAI 2026.

---

### Author Response · Authors · 2025-11-28
**Summary of Response**

We thank the reviewers for their insightful comments and positive assessment. We are encouraged that the reviewers unanimously recognized the critical motivation of our work—addressing the irreversibility of GUI actions [R1, R3]—and the novelty of the GAIA Data Flywheel in capturing realistic error distributions compared to heuristic methods [R1, R2, R4]. Reviewers also highlighted the practical value of our Intuitive Critic Model (ICM) in enabling efficient Test-Time Scaling (TTS) [R2, R3] and the comprehensive empirical gains achieved across diverse open-source and closed-source agents [R1, R2, R4].

In this revision, we have meticulously addressed the reviewers' questions to further strengthen the paper's contribution. We have clarified key concepts within the framework and provided a broader comparison with existing work to better contextualize our contribution. We added extensive experiments to further verify the system's robustness, generalization, and performance boundaries, while also providing proof of efficiency. Additionally, we included a detailed discussion on the role of Reinforcement Learning (RL) in critic training, clarifying why our Intuitive Critic (based on SFT) currently outperforms Reasoning Critics (based on RL) by injecting essential discriminative knowledge—thereby establishing a foundational reward model for future actor-critic RL cycles. We believe these comprehensive revisions and clarifications solidify GAIA as a fundamental contribution to scalable and self-improving GUI agents.

---

### Author Response · Authors · 2025-12-02
**Global Response [Update of PDF & Point-by-Point Reply]**

We would like to express our sincere gratitude to the Reviewers, Area Chairs, and Program Chairs for their time and dedicated effort. We have carefully studied all comments and have provided detailed point-by-point responses under each Reviewer's section.

Our approach not only achieved state-of-the-art performance, but also gained recognition in the following aspects:
* Novelty and necessity: "practical and necessary solution (R_q4WG)"; "addresses a key bottleneck for GUI agents (R_dudK)"; "addresses a clear challenge in GUI agents (R_JEcH)"; "innovative Data Flywheel System (R_ZSGR)".
* Method: "data flywheel is crucial (R_q4WG)"; "overall pipeline and roles of ICM/ICM-r2 are clear (R_dudK)"; "clearly structured (R_JEcH)"; "well-designed (R_ZSGR)".
* Experiment: "comprehensive (R_q4WG)"; "performance gain across various models and benchmarks (R_q4WG)"; "solid empirical study across multiple agents and benchmarks (R_dudK)"; "gains generalize (R_dudK)"; "illustrate the generality (R_JEcH)"; "demonstrating the practical utility of the approach (R_ZSGR)".

In response to the reviewers' questions, we have clarified the following key issues and detailed in the revision (highlighted in blue):
* Comparison with existing GUI critics. We validated the superiority of GAIA's realistic negative sampling over heuristic approaches (UI-Genie-RM in Table 4) and demonstrated that ICMs outperform reasoning-based critics in both judgment accuracy and inference efficiency for test-time scaling (Table 5).
* Definition of positive and negative samples in GAIA. We clarified the labeling criteria used in our dataset construction, specifically detailing how the system accounts for subtle but tolerable variations in action parameters.
* Benefits of the data flywheel. We systematically demonstrated the efficacy of the flywheel mechanism, detailing the consistent performance improvements of GAIA-driven ICM and ICM-r2 across three datasets, two experimental settings, and five diverse base agents, validating the method's universality and iterative scaling trend.
* Clarification of theoretical insight (SFT vs. RL). We showed that our ICMs (83.19% and 83.56%) significantly surpass RL-based reasoning critics (70.82% of RCM and 69.20% of GUI-Critic-R1) in accuracy and single-token efficiency, establishing the intuitive model as the optimal choice for current scaling and a necessary foundation for future research.

We also conducted additional experiments to address the following questions:
* Time complexity analysis. We verified the efficiency of our test-time scaling approach. Leveraging parallel optimization, we demonstrate that increasing rollouts (N=1 to 8) raises latency marginally (~1.49s to ~3.25s) while boosting success rate by up to 17.3%.
* Out-of-distribution generalization: We highlighted adaptability across two dimensions. 1) Our ICM (trained exclusively on high-level prompts) effectively improves agents regardless of whether they prompt on high-level goals or low-level single-step instructions, and further generalizes to the unseen ScreenSpotV2 dataset despite not encountering these specific distributions during training. 2) We conducted online testing on the AndroidWorld benchmark. Our ICM-driven agent achieved an SOTA 28.0% success rate, outperforming the reasoning-based GUI-Critic-R1, proving robustness even in high-entropy live settings.
* Comparison with additional baselines. We demonstrated that ICM (83%+) significantly outperforms generalist VLMs (56.11% of zero-shot Qwen 2.5 VL) in judgment accuracy. Furthermore, we proved that ICM-guided scaling (47.8% and 50.2% SR) dominates naive $N$ rollout majority voting (45.3%) on GUI-Odyssey, confirming the necessity of a learned critic.

In addition, we noticed some errors made by the reviewers.
* Task difficulty (R_q4WG): We clarified that reducing visual history actually increases judgment difficulty due to lower observability; this is a deliberate design choice for TTS efficiency, not a simplification to lower the baseline.
* Contribution nature (R_JEcH): GAIA is not merely engineering; it fundamentally solves the data scarcity challenge for GUI critics and establishes the necessary data-driven foundation (Intuitive Critic as Reward Model) for future end-to-end Actor-Critic training.
* Efficiency comparison (R_ZSGR): The concern regarding efficiency stems from a misunderstanding of the GTA1 workflow. We clarified that our parallelized process is highly streamlined to ensure low latency. Furthermore, we emphasized that the original paper already provides comprehensive experimental evidence confirming the superiority of our Intuitive Critic over reasoning-based alternatives.

We sincerely hope that the above clarifications and supplementary experiments have demonstrated the effectiveness and superiority of the proposed method, and that our modifications have improved the quality of our work and paper.

Best regards,

Authors of Submission 281

---

### Meta-Review · Area_Chair_tyxM · 2026-01-05

**Summary:**

Summary:
This paper presents GAIA, a GUI Action Critic’s Data Flywheel System designed to reduce irreversible errors in GUI agents by introducing a dedicated Intuitive Critic Model (ICM) that evaluates actions before execution. GAIA trains the ICM as a binary classifier to judge whether a candidate action is correct based on the current screen, instruction, action history, and proposed operation, and uses it at test time to select the best action from multiple stochastic rollouts, enabling effective Test-Time Scaling. The framework operates in an iterative flywheel. Experiments demonstrate consistent improvements in action accuracy, planning, and grounding.

Strengths:
1. The paper addresses a critical challenge for GUI agents: irreversibility of erroneous actions.
2. The flywheel-based system is well motivated, clearly structured, and seems to boost performance.
3. The empirical evaluation is clear and solid, and it includes both open-source and closed-source agents.
4. Technical details are clearly presented.

Weaknesses (initial concerns from reviewers):
1. The main contribution, the flywheel, appears to stall after a single turn.
2. Unclear whether improvement came from simplifying the inputs.
3. Unclear whether and how wrong labels may affect the performance.
4. Unfair comparison: The RCM is trained on a 30k subset while ICM/ICM-r2 leverage larger flywheel data
5. There is no latency or cost analysis.
6. Lack of generalization evaluation: training data were drawn from the same benchmarks used for evaluation.
7. Missing baselines: GUI-Critic-R1 and the pretrained Qwen-2.5-VL.
8. Lack of online benchmark evaluation.

**Reviewer Concerns:**

The authors provided additional results, including more baselines and evaluation on the AndroidWorld benchmark for online testing.  In particular, concerns 1, 2, 3, 5, 6, 7, and 8 have been addressed.

However, there are a few remaining main concerns:
1. The data used for training RCM was less than the proposed method. While it is a typical training setting as in prior work, the question about the fairness of the comparison remains unaddressed here, when the proposed method had access to more data.
2. The main finding was that “we showed that our ICMs (83.19% and 83.56%) significantly surpass RL-based reasoning critics (70.82% of RCM and 69.20% of GUI-Critic-R1) in accuracy and single-token efficiency, establishing the intuitive model as the optimal choice for current scaling and a necessary foundation for future research”. While valuable, this was not a theoretical insight as claimed by the authors during rebuttal. Rather, this was an empirical observation. Considering the first concern about the fairness of the comparison, it may also require a more rigorous validation.
3. While the addition of the experiment on an online benchmark (AndroidWorld) is valuable, it is not in the original submission, and thus, the current submission would need a significant revision.

**Reviewer Scores:**

I think the reviewers will maintain their scores if they had been able to participate fully in the discussion. Specifically,

For q4WG, the unfair comparison was not fully addressed.

For dudK, the original score was already 6. The rebuttal did not provide strong arguments that suggest a higher degree of novelty and potential impact.

For JEcH, the weakness of the limited contribution of the methodology itself was true and not fully addressed by the rebuttal.

For ZSGR, the unfair comparison (less data for RCM) was not fully addressed. The complaint about the lack of online evaluation was partially addressed by the new experiment on AndroidWorld. But there was no discussion of this new experiment in the current submission. There are also no detailed results on this benchmark. Thus, it would require further major revision.

---

### Decision · Program_Chairs · 2026-01-26

Reject